behaviour

acoustic communication, Cape fur seal, colonial life, vocal repertoire, acoustic partitioning

**Author for correspondence:**
Mathilde Martin
e-mail: mathilde.martin@u-psud.fr

# Vocal repertoire, micro-geographical variation and within-species acoustic partitioning in a highly colonial pinniped, the Cape fur seal

Mathilde Martin[1,2], Tess Gridley[2,3],
Simon Harvey Elwen[2,3] and Isabelle Charrier[1]

[1]Equipe Communications Acoustiques, Neuro-PSI, CNRS UMR 9197, Université Paris Saclay, Orsay, France
[2]Sea Search Research and Conservation NPC, 4 Bath Road, Muizenberg, Cape Town 7945, South Africa
[3]Department of Botany and Zoology, Faculty of Science, Stellenbosch University, Stellenbosch 7605, South Africa

MM, 0000-0002-9314-4729; IC, 0000-0003-4873-2342

Communication is fundamental for the survival of animal species as signals are involved in many social interactions (mate selection, parental care, collective behaviours). The acoustic channel is an important modality used by birds and mammals to reliably exchange information among individuals. In group-living species, the propagation of vocal signals is limited due to the density of individuals and the background noise. Vocal exchanges are, therefore, challenging. This study is the first investigation into the acoustic communication system of the Cape fur seal (CFS), one of the most colonial mammals with breeding colonies of hundreds of thousands of individuals. We described the acoustic features and social function of five in-air call types from data collected at two colonies. Intra-species variations in these vocalizations highlight a potential ability to convey information about the age and/or sex of the emitter. Using two classification methods, we found that the five call types have distinguishable frequency features and occupy distinct acoustic niches indicating acoustic partitioning in the repertoire. The CFS vocalizations appear to contain characteristics advantageous for discrimination among individuals, which could enhance social interactions in their

noisy and confusing acoustic environment. This study provides a basis for our understanding of the CFS acoustic communication system.

## 1. Introduction

Communication is fundamental for the reproduction and survival of many animal species [1]. Sensory signals (acoustic, visual, olfactory or tactile) are involved in many social interactions and serve for recognition of individuals, mate selection, parental care or coordination of collective behaviours (i.e. foraging and anti-predator strategy) [2]. To fully understand the functioning of a social system, it is essential to know how information is exchanged between individuals. Communication is rarely a uni-modal process but involves several sensory modalities. Because the acoustic channel can be very efficient and reliable in many cases, it is an important modality used by birds and mammals in many social contexts. Indeed, depending on their characteristics and the environment they are produced in, acoustic signals can be propagated over long distances, can be easily localized and be used in the dark. Sound is, therefore, one of the most efficient sensory cues for both short- and long-range communication [3]. Nevertheless, information exchange between individuals via acoustic signals is subject to multiple constraints resulting from the social structure and the habitat of the species. In group-living animals and particularly colonial species, individuals are involved in a wide range of interactions across various social contexts. The effectiveness of vocal exchanges is, therefore, limited by factors such as the density of individuals and environmental constraints such as high background noise which limits the propagation of acoustic signals [4–6]. This highlights the need for an adapted and efficient communication system allowing individuals to establish and maintain reliable inter-individual communication exchanges. This is crucial for the fundamental biological processes of a species (such as mate selection, reproduction and parental care).

Communication is generally defined as a dyadic process in which information is transmitted from an emitter to a receiver through a signal in which information is encoded [7]. But for many group-living species, we have to consider communication as a network of exchanges among multiple individuals constantly in close contact [8]. Based on the 'social complexity hypothesis for communication' [9], the coding–decoding mechanisms are likely to be more elaborate in a complex social and communicative system. This includes the physical characteristics of the signals produced, the quantity of information they convey, the specific properties of features encoding information but also the cognitive abilities of the individuals to analyse the received signal. Evidence for a positive correlation between social and vocal complexity is found in several taxa: birds [10], bats [11], rodents [12], primates [13], mongoose species [14], whales [15] and phocid seals [16,17]. For species living in dense groups, vocal complexity usually causes an increase in the vocal repertoire size. However, other mechanisms can be involved such as the use of more elaborate acoustic signals (i.e. conveying information about identity, sex, social position, age), the use of discrete or gradual calls, the context-dependent nature of calls or the modulations they exhibit [14].

Here, we investigate the acoustic communication system of the Cape fur seal (CFS), *Arctocephalus pusillus pusillus*. This otariid (eared seal) species is particularly interesting to study from an acoustic perspective because of the extreme density of colonies and their social structure during the breeding season, which requires multiple exchanges among individuals in diverse social contexts. The CFS is one of the most colonial mammals in the world with colonies up to hundreds of thousands of individuals during the breeding time. As with all fur seal species, CFS colonies are organized in harems with a high degree of polygyny [18] and various interactions among adults of both sexes. With regard to males, socially mature males (bulls) compete for females and territories while subadult males are chased away from harems by these bulls. Females also engage in agonistic interactions, mostly with other females, to protect their young and compete for space in the colony. In addition, the repeated mother's foraging trips at sea during the lactation period and the absence of allo-suckling requires a reliable system of communication between the mother and her offspring, at both short and long distances during reunion contexts [19]. Empirical study of acoustic communication by CFSs is non-existent, although, as with other pinnipeds (earless seals, fur seals, sea lions and walruses), it is probably the primary communication channel used by this species [20].

Our first aim was to identify all call types produced by individuals of different age and sex categories during the breeding season. We intended to describe their acoustic characteristics and the behavioural contexts of call production, to help deduce their social function. Most studies on the vocal repertoire

of pinnipeds are associated with the investigation of individuality in vocalizations [21–29]. However, here our initial investigations are focused at a larger scale and we explore acoustic differences in vocalizations produced by individuals of different social roles in the colony (age class, sex, breeder/non-breeder, territorial/non-territorial). For each call type, we assessed whether the signal contained unique acoustic features which could potentially be used to convey information about the age and/or sex of the emitter.

Our second aim was to investigate the relative organization of CFSs' call types in their acoustic space. In a CFS colony, individuals of any age, sex, social or reproductive status are aggregated in space. Thousands of seals vocalize at the same time and individuals are constantly exposed to sounds from several emitters (i.e. it is an extremely noisy environment with a high risk of auditory confusion among conspecifics). In such a constrained environment, it is particularly challenging for emitters to produce signals in which information is surely and reliably conveyed. From the receiver perspective, reducing uncertainty regarding the identity (age, sex, social or reproductive status) of a signalling conspecific and decoding the information contained in the received signal is crucial to respond to a given situation [11]. We investigate how CFSs may overcome these constraints and minimize interference among conspecifics. We based our framework on the 'acoustic niche hypothesis' formulated by Krause in 1993 [30] stating that species in a community may partition their acoustic signals in three-dimensional niches (time, acoustic frequency and space) to avoid overlap and improve their intraspecific communication. According to Krause, a niche can be defined as a channel or a space 'in the frequency spectrum and/or time slot that is occupied by no other at that particular moment'. We hypothesized that CFSs' vocalizations might occupy different areas of their acoustic space to facilitate call discrimination. This could allow individuals to quickly identify the social role of a caller (i.e. the age, sex or social position of a calling conspecific). Evidence of acoustic partitioning has been found in several multi-species communities of invertebrates [31–34] and vertebrates [35–38], but to our knowledge, it has never been investigated at the intra-species level.

Lastly, our third aim was to investigate geographical variation in calls by comparing CFS vocalizations from two breeding colonies that are 150 km apart and exhibit different topographical traits. Geographical variations in vocalizations have been widely found in pinnipeds (for review see [39]) including otariid species [40]. Vocal variability within a species can occur at two different scales: micro-geographical variations are found between interbreeding populations while geographically isolated populations may exhibit macro-geographical differences [41]. Geographical variation has been demonstrated at the level of the vocal repertoire (number and types of vocalizations different among study locations), for instance in Weddell seals [42–44] but also at the level of a call type, with characteristics that differed between geographical areas (for instance in bearded seal [45–47]). In otariids, such geographical variations in repertoire composition and its characteristics have been less well investigated, however, examples of both micro- and macro-geographical variation have been found in Australian sea lion (*Neophoca cinerea*) male barks [48,49] and in both mothers' and pups' contact calls of South American sea lions (*Otaria flavescens*) [50]. Here, we investigated geographical variation in both CFS vocal repertoire and the calls' acoustic features to assess the possible effect of the environment on their vocalization characteristics.

# 2. Material and methods

## 2.1. Animals and recording procedure

The CFS breeding season occurs from late October to early January each year [51,52]. At the start of the breeding season, socially mature males establish territories and form harems monopolizing 10–30 females [53]. CFS males become sexually mature at around 5 years (i.e. subadults), but they only reach the social maturity when they are 9–13 years old [54]. We determined the males' age class (subadult versus adult) based on their physical characteristics and not on their abilities to hold harems as some adult males do not hold harems. Adult males are much larger than subadult males, and they also show specific physical features: enlarged neck and shoulders, mane with longer guard hair around the neck and shoulders [54]. Subadult males are found on the edge of the colony, whereas most mature males hold their harem through aggressive behaviour including vocalizations. Females become sexually mature earlier than males, between 3 and 6 years old [53]. The subadult age class, as found in males, does not occur in females as they shift directly from the juvenile stage (non-reproductive—not considered in this study) to the adult stage. Females give birth to a single pup each year and exclusively nurse their young. Like all otariid species, the lactation period is long (weaning

occurs around 9–11 months in CFS [19]) and interspersed with maternal attendance periods on shore and foraging trips at sea. In CFS, the mother's first departure to sea occurs 6 days after parturition [19] and females are absent for approximately 70% of the time during the lactation period [55].

Acoustic recordings were conducted during the breeding season 2019–2020 at two CFS colonies located on the Namibian coast: Pelican Point (PP) (25°52.2′ S, 14°26.6′ E) and Cape Cross (CC) (21°46.5′ S, 13°57.0′ E) (see electronic supplementary material, figure S1). PP is a mainland colony that developed relatively recently. Initially described as a non-breeding colony in 1988 [56], pup production has greatly increased during recent years with up to 12 000 at the last aerial census in 2011 and the entire population now reaches 80 000 seals during the breeding season (N. Dreyer 2021, personal communication). PP is a dynamic sandy peninsula with uniform and flat topography. Most breeding occurs along the leeward edge of the peninsular, which experiences less wave exposure (i.e. little or no wave noise) compared with the western side which is exposed to high waves and swell. Data collection took place at the leeward groups. CC is the world's largest breeding colony of CFSs with about 210 000 individuals in total each year during the breeding season (MFMR [57]) and up to 54 000 pups estimated from aerial census in 2003 [58]. The CC colony expands across a wide rocky bay with steep topography. The coastline is directly exposed to the swell, and when strong, it can generate loud wave noise. The two colonies are separated by about 150 km.

Fur seals' vocalizations were recorded using a Sennheiser ME67 directional shotgun microphone (frequency range: 40–20 000 Hz ± 2.5 dB) using a 44.1 kHz sampling rate and connected to a two-channel NAGRA LB or Roland R26 digital audio recorder. Recordings were made after a 15 min period of habituation to ensure that individuals behaved naturally and to avoid any disturbance caused by our presence. While recording, the microphone was facing on-axis directly towards the vocalizing animal, and the experimenter was describing the identity, sex, age class and the behaviour of the animal producing vocalizations in a lapel microphone connected to the second channel. The distance between the calling animal and the microphone ranged from 0.5 to 6 m while recording (visual estimation).

Pups and adults of both sexes were recorded at PP, as well as subadult males. At CC, no adult males were available to record, but all other age/sex classes were documented. Variation in pup vocal behaviour was investigated in more detail by dividing the pup age class into three age sub-categories. We estimated the age of pups as follows: very small pups with umbilical cord remains were categorized as 'less than two weeks old'; intermediate size pups without umbilical cord remains and recorded at the beginning of December were considered as 'one month old' age class. The class 'two to four months old' included only large size pups showing some signs of moulting which were recorded in February–March. At PP only, some pups were briefly caught to sex them when it was possible (i.e. mother at sea or asleep, pup approaching observers without stress). Many of our sampled pups were marked with hair dye (hair dye: Blonde high-light kit, ©Kair) at a very young age (less than two weeks) so that they could be re-recorded over the four-month period. Males, females and pups were recorded in various contexts: mothers and pups searching for each other, agonistic interactions between individuals and mating behaviours.

## 2.2. Acoustic measurements

Recordings were converted from stereo to mono and resampled at 22.05 kHz as none of the calls' frequency exceed 10 kHz. Acoustic analyses were performed using Avisoft SAS Lab Pro (R. Specht, v. 5.2.14, Avisoft Bioacoustics, Berlin, Germany) and spectrograms were calculated with a 1024-point fast Fourier transform (FFT), 75% overlap and a Hamming window (frequency resolution = 21.5 Hz). Only good-quality calls with low background noise (signal to noise ratio visually inspected on spectrograms) and no overlap with other vocalizations were selected for further analysis. Selected calls were individually high-pass filtered at 100 Hz. Signals were categorized into different call types based on the behavioural context of production and on the previous knowledge of other fur seal species: South American fur seal (*Arctocephalus australis*) [59,60], Australian fur seal (*Arctocephalus pusillus doriferus*) [61], Subantarctic fur seal (*Arctocephalus tropicalis*) [62] and Northern fur seal (*Callorhinus ursinus*) [63]. Five different call types were identified and were based on the social context (most calls being already described in the literature, Insley *et al.* [20]): pup attraction call (PAC), female attraction call (FAC), bark, growl and long bark. A maximum of 10 calls of the same call type (or 10 sequences for barks) per individual were included in the analysis. We measured a set of nine parameters common to all call types: total duration of the call (*Dur*; ms), fundamental frequency (*f0*; Hz; measured with the harmonic cursor function), the frequency value of the first, second and third energy peak (*Fmax1*, *Fmax2*, *Fmax3*; Hz), quartiles of energy spectrum (*Q25*, *Q50*, *Q75*; Hz) and the

frequency bandwidth within which the energy falls within 12 dB of the first peak (*Bdw12*; Hz). The 10th variable corresponded to the percentage of energy below 500 Hz for adults' calls (*Ebelow500*; %) and below 2000 Hz for pups (*Ebelow2000*; %). The total duration was measured on the waveform with a cursor precision of 1 ms and spectral parameters (*f0*, *Fmax1*, *Fmax2*, *Fmax3*, *Q25*, *Q50*, *Q75*, *Ebelow500/Ebelow2000* and *Bdw12*) were measured from the averaged energy spectrum (Hamming window, frequency resolution = 21.5 Hz, frequency range: 0–5000 Hz). As barks were always produced in sequence, measurements were thus performed on five barks randomly chosen from each bark sequence. In addition to the 10 variables described above, we also measured the inter-bark duration (*InterbarkDur*; duration of the silence between the end of the measured bark and the beginning of the next one) for each of the barks and the duration of the bark sequence (*DurSeq*; ms).

Other additional acoustic variables were also measured to exhaustively describe the vocal repertoire but were not included in the multivariate analyses. This was necessary to describe characteristics specific to a given type of call but not present in all calls of that type. For instance, in PACs, some females produce a noisy part whose temporal pattern consists mainly of a fast amplitude modulation (AM). We determined the fast AM proportion (*AMprop*; %) which is the ratio between the duration of the fast AM part and the total duration of the call (×100) and the rate of the AM (*AMrate*; Hz), using the pulse train analysis function in Avisoft. In addition, some pup calls include a period of bleating or quavering (i.e. fast frequency modulation) [64] so we measured the bleating rate (*Brate*; Hz) and the bleating proportion occurring in the call (ratio between the duration of the bleating part and the total duration of the call × 100) (*Bprop*; %). *Bperc* (%) represents the percentage of pups showing a bleating part in their analysed calls. Growls often show fast AMs and, in some cases, can be composed of two parts separated by a short silence. We measured the AM pulse rate for the part 1 (*Pulse1*; Hz) and the part 2 (*Pulse2*; Hz) and we also considered the duration of the first and second part (*Dur1* and *Dur2*; ms) as well as the duration of the silence between the two parts (*Silence*; ms).

## 2.3. Statistical analysis

All statistical analyses were carried out using RStudio v. 1.2.5042 [65]. As an initial test, the set of nine variables common to all call types (*Dur*, *f0*, *Fmax1*, *Fmax2*, *Fmax3*, *Q25*, *Q50*, *Q75* and *Bdw12*) were tested for multicollinearity at the level of 0.8 [66] and the variable *Q50* was removed from the analysis because of its high correlation with the other spectral variables.

Our general approach to addressing the research questions was based on random forest (RF) algorithms to investigate fine-scale differences between vocalizations across multiple groups (i.e. call type by age, gender or study site). The RF is a recent and robust classification method based on averaging multiple decisions trees [67]. Using a bootstrap process, RF predicts the membership of the data (i.e. vocalizations) to a given group for each tree constructed. The indicator of precision gives the number of correct predictions per group and is compared with the prediction expected by chance (if the classification was randomly performed) which is calculated as: number of calls in the group/total number of calls × 100. In addition, the global accuracy of precision is calculated as a proportion of the total correct classification from an RF procedure. Random forest algorithms were conducted using the package *randomForest* [68]. We set the number of trees to be grown at 1000 because the global error rate generally stabilized from this value. The number of variables to be selected at each node was set at the square root of the total number of variables included in the RF, which corresponds to the value suggested by the package. To equalize the role of each class in the categorization and to avoid the over-representation of the biggest classes, we used the 'Balanced Random Forest' algorithm described by Chen *et al.* [69]. This procedure built trees with the same sample size for each group every time (i.e. the number of calls in the smallest group). In addition, the contribution of each variable to the RF classification was assessed through inspection of the Gini index values. All balanced RF procedures mentioned below were performed according to this general method.

### 2.3.1. Vocal repertoire and variations among sex and age classes

Acoustic features of all call types (from both adults and pups) were described using mean and standard deviation values of each of the measured variables (in general and then by age class for pups' calls). Our classification in call types was tested using a balanced RF algorithm separately for PP and CC datasets. RF were performed with the eight variables common to the five call types (*Dur*, *f0*, *Fmax1*, *Fmax2*, *Fmax3*, *Q25*, *Q75*, *Bdw12*) and each tree was built with 17 calls randomly selected from each of the call types for PP and 27 calls for CC (i.e. number of good-quality, complete calls in the smallest call-type category).

Other investigations were then carried out to describe the variations in one call type according to the sex or age class of individuals: pups' calls characteristics were investigated by age class and barks from adults were compared by sex and age (i.e. comparisons between females, adult males and subadult males). First, the differentiation of pup calls with age was examined for both colonies using a balanced RF (using the nine acoustic variables measured on FAC and each tree built with 48 calls for PP and 67 calls for CC). A linear discriminant analysis (LDA) was also performed with the same variables to visualize the separation among age classes for both study sites. Each acoustic parameter was also tested using linear mixed effects models (LME) run with the R package *nlme* [70]. Models were built with each acoustic parameter as the response variable, the age was set as 'fixed effect' and pup identity was defined as a 'random effect' to account for the fact that the dataset included several calls per individual. Acoustic parameters were log-transformed to reduce the non-homogeneity of variances and residuals were inspected for normality and homoscedasticity. LME could not be performed on bleating variables due to small sample sizes. Using a subset of the data from PP, we compared sex differences between very young pups, i.e. those less than two weeks (81 females and 34 males), again using the balanced RF (nine acoustic variables and each tree built with 34 calls) and a Wilcoxon rank-test to investigate differences between acoustic variables (as our data do not have normal distributions and homogeneous variances).

Secondly, adult barks were compared among sex and age class: males' and females' barks (for both PP and CC) as well as adult males' and subadult males' barks (for PP only as adult males were not recorded at CC). Analyses were carried out with a balanced RF for both colonies using 10 variables (*Dur, f0, Fmax1, Fmax2, Fmax3, Q25, Q75, Ebelow500, Bdw12* and *InterbarkDur*). As *Dur_seq* is relative to a sequence of barks and not to a bark, this variable was excluded from RF. Trees were built with the same sample size for each group (i.e. the size of the smallest group): $n = 150$ barks for PP and $n = 135$ barks for CC. A LDA was also performed with the same variables and we plotted barks from females, subadult males and adult males in a two-dimensional space to assess visually their distinctiveness. Wilcoxon rank-tests were conducted two-by-two (adult males versus subadult males, adult males versus females and subadult males versus females) to test for differences in the 10 acoustic variables cited above plus the variable *Dur_seq* (relating to sequences rather than barks). We adjusted the level of significance using the sequential Bonferroni adjustment [71] for analyses on barks from PP and all results retained significance when $p < 0.017$ (i.e. $0.05/3$).

### 2.3.2. Acoustic partitioning

To investigate the relative organization of the CFS repertoire in the acoustic space, we performed a LDA separately on the whole PP and CC datasets using the eight acoustic variables that the calls have in common (*Dur, f0, Fmax1, Fmax2, Fmax3, Q25, Q75* and *Bdw12)*. The resulting LDA plot for the different call types (PAC, FAC, bark, growl and long bark) provides a visual two-dimensional representation of call distinctiveness, and thus call separation in the acoustic space.

### 2.3.3. Micro-geographical variations in vocalizations

Geographical differences in vocalizations between our two study locations were investigated using a balanced RF for each call type. We assessed whether the same type could be correctly classified according to their original recording colony: PP or CC. Again, classification was supplemented by a series of Wilcoxon tests performed on each measured variable to investigate differences in acoustic features. As no adult males were recorded at CC, only subadults were compared between the two colonies.

# 3. Results

## 3.1. Vocal repertoire and variations among sex and age classes

A total of 39 h (32 h from PP recorded over 59 days and 7 h from CC recorded over 5 days) were analysed. A total of 3825 good-quality calls (3038 at PP and 787 at CC) recorded from 362 individuals (67 subadult and adult males, 168 females and 127 pups from both colonies) were included in the analysis. Based on the literature, audio-visual inspection of the spectrograms and the social context of their production, vocalizations were divided into five call types (figure 1, see electronic supplementary material, audio S1), with four recorded in both colonies: FAC, PAC, bark and growl (representing 23%, 20%, 55% and 2% of the calls analysed). The long bark was the least common call

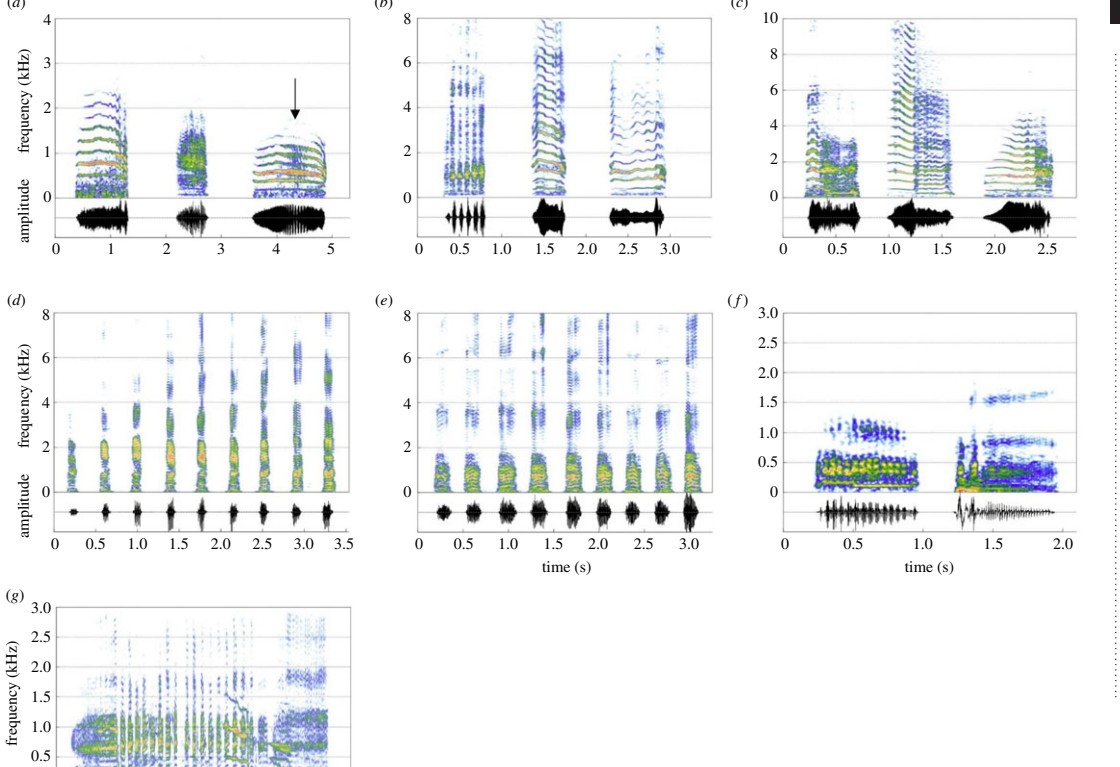

**Figure 1.** Spectrograms of Cape fur seal call types. (*a*): PAC from three different females: clear harmonic structure, noisy call and harmonic structure with a fast AM part (indicated by the arrow), (*b*): FAC from three pups at three different ages: (from left to right) less than two weeks, one month and two to four months old, (*c*): FAC from one pup (P25) at three age classes: less than two weeks, one month, two–four months old, (*d*): female barks, (*e*): male barks, (*f*): soft and loud growls, (*g*): a sequence of male barks starting and ending by a long bark. Spectrogram (Hamming window size: 1024 pts, 90% overlap) generated using Seewave [72]. For a better visualization, the different call types are plotted on different axes.

type (representing 0.5% of the calls analysed) and recorded at PP only. The global balanced RF algorithms performed on PP and CC datasets reported indicator of precision values all higher than expected by chance (figure 2) with a global accuracy of prediction of 99% for PP and 97% for CC. The classification of vocalizations into five different call types is, therefore, supported with high confidence.

*PAC* (figure 1*a*; table 1) is a vocalization produced by females to exchange with their pup at long distance in a context of reunion after a foraging trip at sea and at short distance when gathered in the colony. The PAC is a relatively long harmonic call (*Dur*: 1118–1128 ± 306–357 ms; representing the mean values for PP and CC followed by their respective s.d. values) composed of a fundamental frequency (*f0*: 264–276 ± 41–39 Hz) and its harmonic series. Half of the energy was concentrated below 1000 Hz (*Q50*: 900–1115 ± 269–315 Hz) and most of the energy was around Fmax1 (*Bdw12*: 15–16 ± 17–14 Hz) (table 1). About 15% of females produced a noisy part showing a fast AM pattern occurring on average for half of the total duration of the call (*AMprop*: 61–59 ± 30–22%) and with an averaged *AMrate* of 53–36 ± 25 Hz (table 1).

*FAC* (figure 1*b,c*; table 1) is the most common vocalization produced by pups and occurs in the same behavioural contexts as the PAC (reunion and close contact). In comparison to the PAC, the FAC was shorter (*Dur*: 637–600 ± 219–195 ms) and higher pitched (*f0*: 364–379 ± 79–72 Hz; *Q50*: 1283–1625 ± 511–646 Hz) (table 1). In general, for all parameters, the standard deviation values were higher than for the PAC, suggesting greater variability. Bleating was observed in 51% (PP) to 39% (CC) of the recorded pups, and when present in calls, it occurred with a long duration (*Bprop*: 85–92 ± 29–17%) and the average bleating rate was around 10 Hz (table 1).

The mean acoustic features of the FAC by pup age classes are shown in table 2. RF algorithms reported a global accuracy of prediction of 68% and 74% for PP and CC, respectively (see electronic

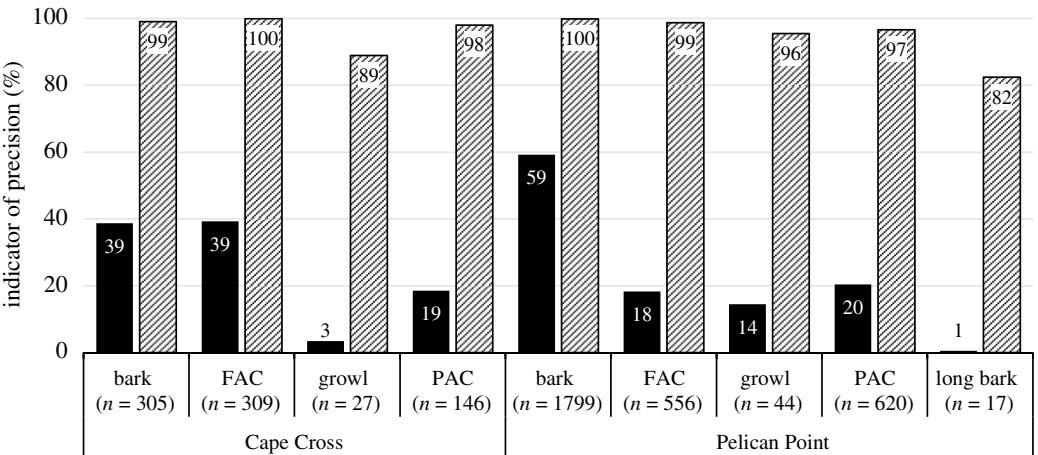

**Figure 2.** Comparison of accuracy of prediction of vocalizations by call type at CC and PP obtained by chance or using the Balanced Random Forest algorithm performed with eight variables (*Dur, f0, Fmax1, Fmax2, Fmax3, Q25, Q75, Bdw12*). Data are pooled by call types, the latter including individuals of all ages and sexes if any.

supplementary material, figure S2). Within the RF analysis, the variables with the greatest predictive power were *Fmax1* and *Dur* for PP (Gini index: 18.21 and 11.60, respectively) and *Ebelow2000* for CC (Gini index: 29.76). Values of the indicator of precision were higher than prediction by chance for all age classes suggesting that FACs vary with age and that the set of acoustic variables allowed us to accurately classify them (see electronic supplementary material, figure S2). The LDAs extracted two linear discriminants with *Dur* and *Ebelow2000* being the variables contributing the most to the total variation (see electronic supplementary material, table S1). The correlation with LD1 was positive for *Dur* and negative for *Ebelow2000* in both colonies. *Dur* was positively correlated at PP and negatively correlated with LD2 at CC while *Ebelow2000* is positively correlated to LD2 in both cases (see electronic supplementary material, table S1). The LDA scatterplots (figure 3) showed that, especially for PP, age classes 'one month old' and 'two to four months old' were the most distinct, with age class 'less than two weeks old' sitting between them. This might suggest a progressive trend with age: when getting older, pups increased the duration of their calls, and the distribution of energy was more evenly distributed among harmonics (figure 1*c*; table 2). Regarding the bleating, the proportion of bleating pups (*BPerc*) decreased with age, and the proportion of bleating within the call (*Bprop*) tended to decrease with age as well, whereas the bleating rate (*Brate*) remained constant over the three age classes (10 Hz—figure 1*b,c*; table 2). LME (results presented in table 2) revealed significant differences for most acoustic parameters with—in general—age class 'two to four months old' being different from the two younger ones.

Differences between sexes were only investigated for very young pups (less than two weeks old, nine females and six males) at PP because too few pups from the two other age classes could be sexed. The RF indicator of precision values was higher than expected by chance (see electronic supplementary material, figure S3) with a global accuracy of prediction of 72%. The variables that contributed most to the classification were *Fmax1*, *Dur* and *Ebelow2000* (Gini index: 6.53, 3.87 and 3.78, respectively). Regarding acoustic parameters, we found that in these very young pups, males showed significantly higher average values for *Fmax1*, *Ebelow2000* and *Bdw12* (see electronic supplementary material, table S2).

*Barks* are comparatively short calls (*Dur* between 116 ± 31 ms for females at PP and 125 ± 36 ms for adult males at PP) always produced in sequences (figure 1*d,e*; table 1). Barks were produced by both male and female adults, during agonistic interactions, territorial defence and mating behaviours. Although less frequent, some females were observed producing barks to exchange with their pup in both colonies. On a few occasions, pups of a few months old were also observed producing barks while playing and fighting with other pups but these were not recorded. The fundamental frequency of barks was low (*f0* between 124 ± 28 Hz for adult males at PP and 164 ± 33 Hz for females at PP) and the energy was widely spread along the spectrum with a small proportion below 500 Hz (*Ebelow500* between 8 ± 3% for subadult males at CC and 24 ± 11% for adult males at PP) (table 1).

**Table 1.** Mean (±s.d.) values of the measured acoustic variables for the Cape fur seal call types and results of the Wilcoxon rank test comparing the means between the two colonies Cape Cross and Pelican Point. Dur, InterbarkDur, Dur1, Dur2 and Silence in ms— DurSeq in s— f0, Fmax1, Fmax2, Fmax3, Q25, Q50, Q75, Bdw12, AMrate, Brate, Pulse1 and Pulse 2 in Hz—AMProp, Bprop and Bperc in %. Significance code: 0 '***' 0.001 '**' 0.01 '*' 0.05 'n.s.' 1. Numbers in parentheses indicate the number of calls when the variable is measurable on part of the total calls. n.a., not applicable as statistic tests could not be performed due to sample size.

| | Nall | Dur | f0 | Fmax1 | Fmax2 | Fmax3 | Q25 | Q50 | Q75 | Ebelow500 | Bdw12 | AMprop | AMrate | | |
|---|---|---|---|---|---|---|---|---|---|---|---|---|---|---|---|
| **PAC.** | | | | | | | | | | | | | | | |
| CC | 146 | 1128 ± 357 | 276 ± 39 | 757 ± 277 | 788 ± 420 | 1010 ± 481 | 732 ± 173 | 1115 ± 315 | 1668 ± 406 | 12 ± 10 | 16 ± 14 | 59 ± 22 (17) | 36.20 ± 24.96 (17) | | |
| PP | 620 | 1118 ± 306 | 264 ± 41 | 667 ± 181 | 583 ± 269 | 747 ± 491 | 619 ± 140 | 900 ± 269 | 1483 ± 452 | 17 ± 10 | 15 ± 17 | 61 ± 30 (93) | 52.74 ± 25.10 (93) | | |
| Wilcoxon CC/PP | | n.s. | ** | *** | *** | *** | *** | *** | *** | *** | * | n.a. | n.a. | | |
| **FAC** | | | | | | | | | | *Ebelow2000* | | *Bprop* | *Brate* | *Bperc* | |
| CC | 309 | 600 ± 195 | 379 ± 72 | 1128 ± 541 | 1198 ± 837 | 1247 ± 982 | 988 ± 306 | 1625 ± 646 | 2560 ± 910 | 62 ± 20 | 26 ± 23 | 92 ± 17 (120) | 10.03 ± 1.66 (120) | 38.83 | |
| PP | 556 | 637 ± 219 | 364 ± 79 | 881 ± 421 | 781 ± 560 | 1117 ± 955 | 772 ± 264 | 1283 ± 511 | 2339 ± 999 | 71 ± 15 | 24 ± 21 | 85 ± 29 (286) | 10.28 ± 1.92 (286) | 51.44 | |
| Wilcoxon CC/PP | | * | ** | *** | *** | * | *** | *** | ** | *** | n.s. | n.a. | n.a. | n.a. | |
| **bark (subadult male)** | | | | | | | | | | | | *InterbarkDur* | *Dur_seq* | | |
| CC | 170 | 118 ± 27 | 130 ± 32 | 754 ± 176 | 886 ± 362 | 1021 ± 509 | 754 ± 104 | 1212 ± 289 | 2051 ± 467 | 8 ± 3 | 55 ± 27 | 209 ± 39 | 10.1 ± 7.5 (34) | | |
| PP | 320 | 120 ± 30 | 128 ± 25 | 642 ± 225 | 630 ± 270 | 565 ± 308 | 577 ± 140 | 832 ± 237 | 1381 ± 500 | 21 ± 10 | 60 ± 41 | 219 ± 42 | 9.1 ± 5.0 (64) | | |
| Wilcoxon CC/PP | | n.s. | n.s. | *** | *** | *** | *** | *** | *** | *** | n.s. | *** | n.s. | | |
| **bark (adult male)** | | | | | | | | | | | | *InterbarkDur* | *Dur_seq* | | |
| CC | 1330 | 125 ± 36 | 124 ± 28 | 558 ± 173 | 550 ± 302 | 583 ± 372 | 539 ± 126 | 868 ± 342 | 1522 ± 568 | 24 ± 11 | 53 ± 32 | 182 ± 53 | 11.8 ± 8.7 (266) | | |
| **bark (female)** | | | | | | | | | | | | *InterbarkDur* | *Dur_seq* | | |
| CC | 135 | 118 ± 31 | 149 ± 32 | 758 ± 265 | 786 ± 338 | 808 ± 443 | 667 ± 143 | 1002 ± 297 | 1538 ± 497 | 16 ± 9 | 58 ± 35 | 226 ± 45 | 7.92 ± 6.46 (27) | | |
| PP | 150 | 116 ± 31 | 164 ± 33 | 660 ± 206 | 611 ± 223 | 578 ± 342 | 582 ± 134 | 857 ± 230 | 1384 ± 558 | 20 ± 9 | 70 ± 38 | 254 ± 57 | 4.78 ± 3.11 (30) | | |
| Wilcoxon CC/PP | | n.s. | *** | * | *** | *** | *** | ** | ** | *** | ** | *** | *** | | |
| **Growl (male)** | | | | | | | | | | | | *Dur1* | *Dur2* | *Pulse1* | *Pulse2* |
| PP | 5 | 545 ± 513 | 118 ± 38 | 292 ± 178 | 374 ± 310 | 365 ± 131 | 314 ± 101 | 542 ± 182 | 1275 ± 755 | 48 ± 14 | 21 ± 13 | 318 ± 102 | 567 ± 619 (2) | 62.80 ± 42.16 | 49.50 ± 6.36 (2) |
| **growl (female)** | | | | | | | | | | | | *Dur1* | *Dur2* | *Pulse1* | *Pulse2 / Silence* |
| CC | 27 | 781 ± 364 | 127 ± 68 | 414 ± 149 | 400 ± 223 | 343 ± 228 | 362 ± 81 | 524 ± 128 | 979 ± 257 | 51 ± 15 | 15 ± 9 | 638 ± 353 (26) | 366 ± 181 (10) | 61.64 ± 24.96 (27) | 91.38 ± 95.41 (8) / 54 ± 35 (3) |
| PP | 40 | 982 ± 1278 | 90 ± 35 | 327 ± 153 | 300 ± 126 | 282 ± 146 | 282 ± 65 | 428 ± 80 | 715 ± 261 | 66 ± 16 | 15 ± 17 | 668 ± 87 | 647 ± 638 (17) | 56.66 ± 30.55 | 45.29 ± 20.89 (17) / 163 ± 123 (2) |
| Wilcoxon CC/PP | | n.s. | ** | * | n.s. | n.s. | *** | ** | *** | *** | n.s. | n.a. | n.a. | n.a. | n.a. / n.a. |
| **long bark** | | | | | | | | | | | | | | | |
| PP | 17 | 1277 ± 737 | 111 ± 38 | 695 ± 142 | 717 ± 278 | 832 ± 342 | 614 ± 106 | 823 ± 151 | 1211 ± 297 | 19 ± 12 | 15 ± 13 | | | | |

**Table 2.** Mean (±s.d.) values of the acoustic variables on female attraction calls depending on age class of pups and age comparisons using a linear mixed effects model at Cape Cross and Pelican Point. Significance level: 0.05. LME could not be performed on bleating variables due to small sample sizes.

| | | less than two weeks old pups | one month old pups | two to four months old pups | significant differences (LME) |
|---|---|---|---|---|---|
| Cape Cross | $N_{calls}$ | 154 | 67 | 88 | |
| | $N_{ind}$ | 19 | 10 | 15 | |
| | Dur (ms) | 568 ± 144 | 537 ± 116 | 702 ± 269 | 1 ≠ 3, 2 ≠ 3 |
| | f0 (Hz) | 379 ± 68 | 376 ± 87 | 381 ± 68 | n.s. |
| | Fmax1 (Hz) | 1177 ± 496 | 884 ± 283 | 1227 ± 695 | 1 ≠ 2, 2 ≠ 3 |
| | Fmax2 (Hz) | 1152 ± 723 | 796 ± 493 | 1585 ± 1049 | 1 ≠ 2, 2 ≠ 3 |
| | Fmax3 (Hz) | 1165 ± 994 | 995 ± 743 | 1583 ± 1039 | 1 ≠ 3, 2 ≠ 3 |
| | Q25 (Hz) | 1005 ± 262 | 785 ± 183 | 1113 ± 369 | 1 ≠ 2, 2 ≠ 3 |
| | Q50 (Hz) | 1574 ± 608 | 1112 ± 330 | 2106 ± 553 | all diff. |
| | Q75 (Hz) | 2549 ± 958 | 1925 ± 868 | 3061 ± 422 | all diff. |
| | Ebelow2000 (%) | 64 ± 20 | 78 ± 12 | 46 ± 13 | 2 ≠ 3 |
| | Bdw12 (Hz) | 24 ± 22 | 30 ± 25 | 27 ± 24 | n.s. |
| | BPerc (%) | 74 | 50 | 20 | n.a. |
| | Bprop (%) | 91 ± 18 | 96 ± 13 | 86 ± 21 | n.a. |
| | Brate (Hz) | 10.3 ± 1.8 | 9.6 ± 1.4 | 10.1 ± 1.2 | n.a. |
| Pelican Point | $N_{calls}$ | 264 | 48 | 145 | |
| | $N_{ind}$ | 37 | 9 | 27 | |
| | Dur (ms) | 660 ± 241 | 558 ± 112 | 667 ± 233 | n.s. |
| | f0 (Hz) | 385 ± 82 | 380 ± 66 | 343 ± 70 | 1 ≠ 3, 2 ≠ 3 |
| | Fmax1 (Hz) | 931 ± 493 | 1005 ± 423 | 826 ± 245 | 1 ≠ 2 |
| | Fmax2 (Hz) | 841 ± 580 | 880 ± 667 | 683 ± 588 | 1 ≠ 3 |
| | Fmax3 (Hz) | 1219 ± 1061 | 1017 ± 677 | 1107 ± 1006 | 1 ≠ 3 |
| | Q25 (Hz) | 841 ± 293 | 870 ± 264 | 718 ± 175 | 1 ≠ 3 |
| | Q50 (Hz) | 1408 ± 561 | 1383 ± 397 | 1165 ± 481 | 1 ≠ 3, 2 ≠ 3 |
| | Q75 (Hz) | 2612 ± 1042 | 2409 ± 826 | 2195 ± 890 | 1 ≠ 3 |
| | Ebelow2000 (%) | 67 ± 15 | 70 ± 12 | 72 ± 14 | 1 ≠ 3 |
| | Bdw12 (Hz) | 23 ± 19 | 18 ± 17 | 29 ± 25 | 1 ≠ 3, 2 ≠ 3 |
| | BPerc (%) | 73 | 67 | 44 | n.a. |
| | Bprop (%) | 87 ± 20 | 92 ± 14 | 69 ± 51 | n.a. |
| | Brate (Hz) | 10.6 ± 1.9 | 10.5 ± 1.5 | 10.1 ± 2.0 | n.a. |

We used a RF to classify barks among adult females, adult males and subadult males at PP (and subadult males versus females at CC). The global accuracies of prediction values were 78% (PP) and 83% (CC) and all values of indicator of precision were higher than expected by chance (figure 4). The variables that contributed most to the classification by RF were f0 and InterbarkDur at PP (Gini index: 62.7 and 62.1, respectively) and Ebelow500 and f0 at CC (Gini index: 33.44 and 21.61, respectively). The LDA extracted, respectively, 93% and 7% of the total variance on LD1 and LD2 (figure 5). For the LDA, barks were mostly separated on LD1 which is positively correlated with the variable InterbarkDur (figure 5, electronic supplementary material, table S3). This indicates subtle differences in classification between RF and LDA. However, using LDA, the three groups were also quite distinct: females produced barks with longer inter-bark interval duration than adult males. Subadult males sit in between with intermediate values of inter-bark interval duration (figure 5).

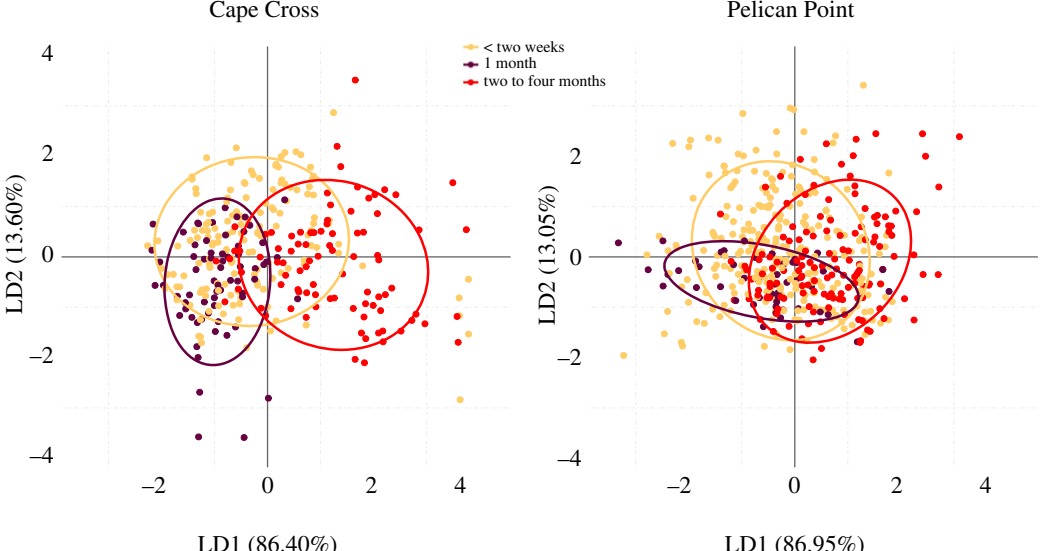

**Figure 3.** Scatterplots of the two linear discriminants resulting from LDAs performed on female attraction calls by age classes at Cape Cross and Pelican Point. LDAs were performed with nine variables (*Dur*, *f0*, *Fmax1*, *Fmax2*, *Fmax3*, *Q25*, *Q75*, *Ebelow2000*, *Bdw12*). *Dur* and *Ebelow2000* are the variables contributing the most to the total variation. Ellipses include 75% of a group's calls.

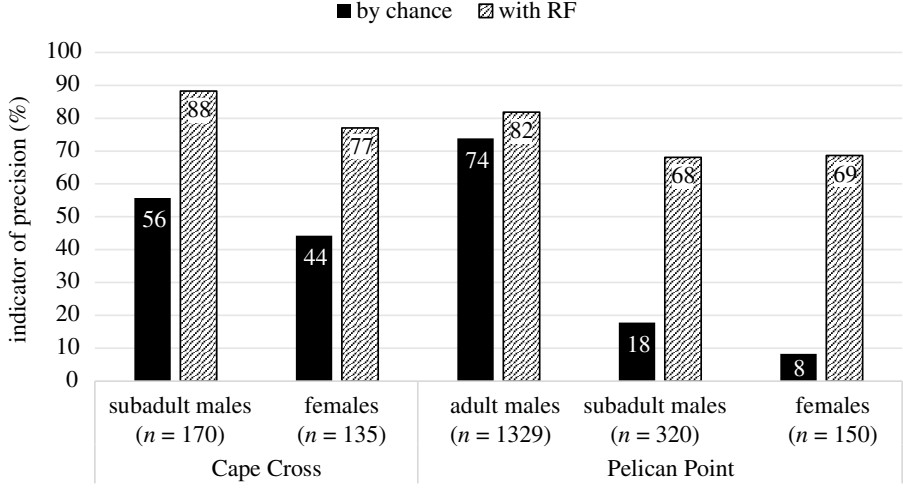

**Figure 4.** Comparison of accuracy of prediction of barks per sex category at CC and PP, and per age class for males at PP, obtained by chance or using the Balanced Random Forest algorithm performed with 10 variables (*Dur*, *f0*, *Fmax1*, *Fmax2*, *Fmax3*, *Q25*, *Q75*, *Ebelow500*, *Bdw12* and *InterbarkDur*).

Differences in barks among adult males, subadult males and adult females were investigated per acoustic variable using a Wilcoxon test. Regarding males, subadults produced higher-pitched barks with a higher inter-bark duration compared with adults (PP) (see electronic supplementary material, table S4). Differences between males and adult females were explored for each geographical location. In both sites, regardless of age, males produced barks with lower fundamental frequency and shorter inter-bark duration. Compared with females, the subadult and adult males at PP displayed longer duration bark sequences (electronic supplementary material, table S4).

*Growls* are pulsed calls (figure 1*f*, table 1) produced by all adults in various social contexts, but mostly during aggressive and agonistic interactions: males fighting for territory and dominance, aggressive behaviour of an adult towards a pup or during agonistic interactions between females. Loud growls were produced during hostile interactions, generally associated with physical demonstration of aggressiveness and mostly related to a high irritation state. Soft growls were rather produced during minor conflicts like females competing for a place to rest. Growls were often associated with barks. Growls are the lowest frequency vocalization (*f0* between $90 \pm 35$ Hz for females at PP and $127 \pm$

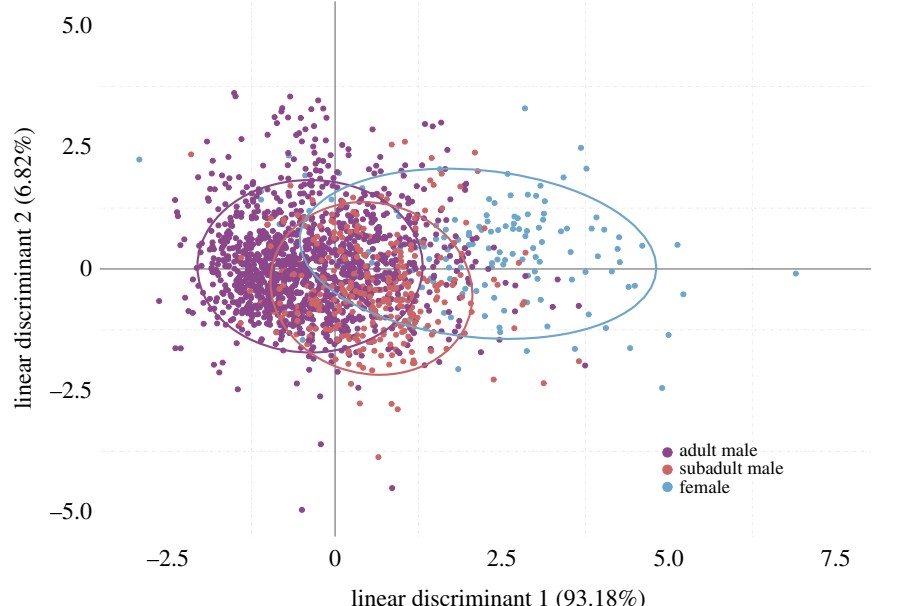

**Figure 5.** Scatterplots of the first two linear discriminants resulting from the LDA performed on adult males' and females' barks and subadult males' barks at Pelican Point with 10 variables (*Dur*, *f0*, *Fmax1*, *Fmax2*, *Fmax3*, *Q25*, *Q75*, *Ebelow500*, *Bdw12*, *InterbarkDur*).

68 Hz for females at CC) we observed with most energy concentrated below 500 Hz (*Ebelow500*: 48 ± 14% for males at PP and −66 ± 16% for females at PP) (table 1). Growl duration was highly variable and growls could be composed of two parts sometimes separated by a silence. Comparison of growls between males and females was impossible due to a small sample size in males (*n* = 5 calls).

Finally, the *long bark* was the least common call type and was only recorded from three males (17 calls in total) at PP. Long barks showed the same spectral characteristics as males' barks but with a much longer duration (*Dur*: 1277 ± 737 ms) (figure 1*g* and table 1). Long barks were produced at the beginning and at the end of the males' bark sequence (figure 1*g*), and there was no specific behavioural context.

## 3.2. Acoustic partitioning

To investigate the organization of the call types in the acoustic space relative to the study site, we performed a LDA on the call repertoire recorded at each location using the eight variables common to these five types of call (*Dur*, *f0*, *Fmax1*, *Fmax2*, *Fmax3*, *Q25*, *Q75* and *Bdw12*). LDAs extracted 99% of the variance for both CC and PP (figure 6). In both cases, the variables *Dur* and *f0* had the highest correlations to the first two linear discriminants (electronic supplementary material, table S5): both were positively correlated with LD1, and *Dur* was positively correlated with LD2 when *f0* was negatively correlated with LD2. Short and low-pitched calls such as growls and barks were thus separated from longer and higher-pitched calls like PAC and FAC on the first axis (LD1). The LD1 thus separated the affiliative calls (FAC and PAC, on the right side) from the other call types, related to agonistic interactions. On the second axis (LD2), growls (top left side) were distinguished from barks (bottom left side) by their longer duration and their lower fundamental frequency (*f0*). The PACs differed from FACs in the same way (PACs, top right side of the plot, were longer in duration and showed a lower *f0* compared with FAC, figure 6).

## 3.3. Micro-geographical variations in vocalizations

Geographical variation in common call types was investigated using a balanced RF algorithm applied to FAC, PAC, female growls, female barks and subadult male barks recorded at each study site (results in figure 7). Global prediction accuracy ranged between 64% and 89% depending on call types and in all cases, the precision indicator values were higher than expected by chance. These results indicate geographical variation in all call types investigated from the repertoire. In all comparisons, call types

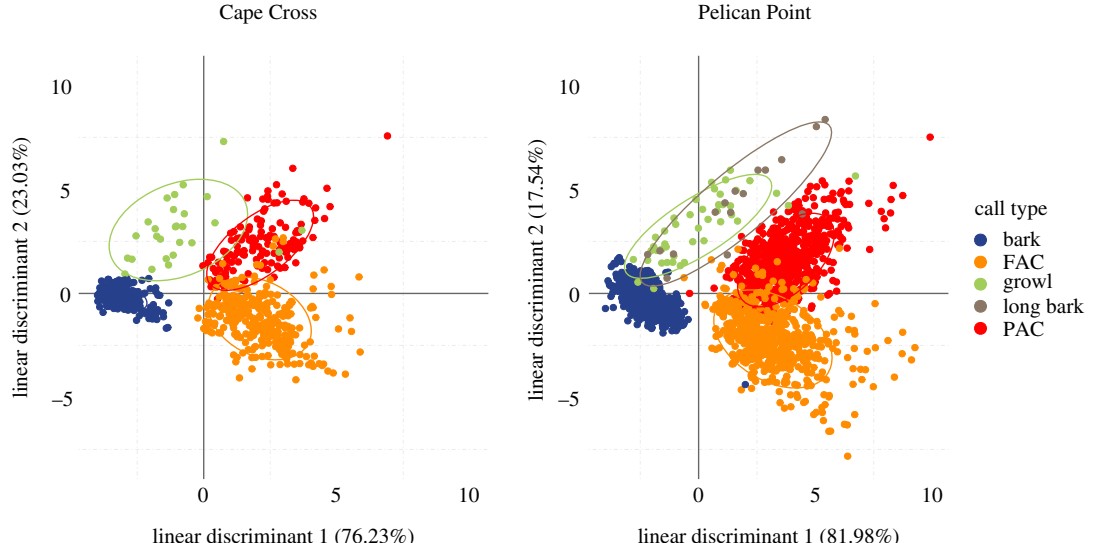

**Figure 6.** Scatterplots of the first two linear discriminants resulting from the LDAs performed on the whole CC and PP datasets with eight variables (*Dur, f0, Fmax1, Fmax2, Fmax3, Q25, Q75, Bdw12*).

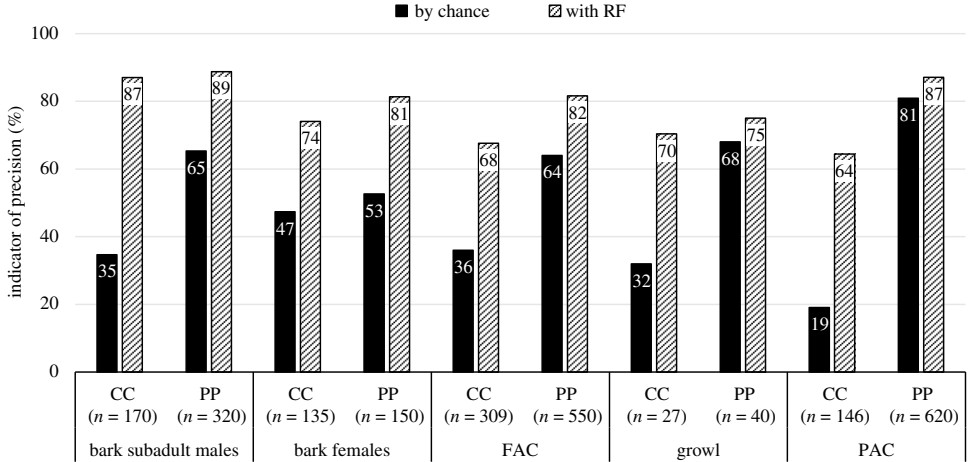

**Figure 7.** Comparison of accuracy of prediction of Cape fur seal call types per breeding colony (Cape Cross, CC and Pelican Point, PP), obtained by chance or using the Balanced Random Forest algorithm performed with nine variables (*Dur, f0, Fmax1, Fmax2, Fmax3, Q25, Q75, Ebelow500* or *Ebelow2000, Bdw12*) and 10 for barks (+ *InterbarkDur*).

were higher-pitched at CC: the frequency of the fundamental (*f0*) and/or all spectral features were higher, and the proportion of energy below 500 Hz (*Ebelow500*) or below 2000 Hz for pups (*Ebelow2000*) were lower. By contrast, call durations did not differ between the two studied colonies (table 1). For barks, females from CC presented sequences significantly longer compared with PP. The inter-bark duration was longer at CC for male barks and longer at PP for female barks (table 1).

# 4. Discussion

## 4.1. Vocal repertoire and variations among sex and age classes

### 4.1.1. Inter-species level

Four of the five call types described in this study were recorded in both colonies: PAC, FAC, bark and growl. Their overall structure and biological function are apparently similar to vocalizations described for other fur seal species, highlighting the fact that the vocal repertoire of otariids is relatively consistent among species [21,23,24,62,73]. However, some species produce additional call types,

mostly involved in agonistic interactions, such as high-pitched calls, full threat calls and submissive calls [24,61,62,73]. We did not find these additional calls types in our study and CFSs seem to only use barks and growls in such agonistic interactions. In spite of a very high-density in breeding colonies, competition among males in this species might be reduced due to the large number of available females. Indeed, males hold harems including on average 28 females, but this can reach up to 66 females [74], whereas the average harem size in other fur seals is about 5–15 females [75]. This could explain why we observed very few physical conflicts (fights) among males and the use of a limited number of agonistic calls in their repertoire. In otariid species, barking is generally restricted to males and the few studies reporting females using barks suggest it is rare [62,76]. In CFS, females produce barks relatively often during agonistic interactions with other females or males. Some mothers were also observed to communicate with their pup through barks. By contrast, we found one vocalization that has never been described before, the long bark. It appeared to be rare and was only recorded at PP from three individuals (one adult and two subadult males). The role of this call remains unknown.

CFS and Australian fur seal (AFS—*Arctocephalus pusillus doriferus*) are two sub-species of *A. pusillus* [77]. The AFS vocal repertoire has been widely described by Tripovich *et al.* [73] and it seems that call types given by the two sub-species are slightly different: in AFS, growls are only produced by females while produced by both sexes in CFS. Submissive calls and guttural threats described in AFS were not found in CFS. Therefore, it appears that since their split (considered as phylogenetically recent, less than 18000 y/a [78,79]) some modifications occurred in the composition of their vocal repertoire, probably due to differences in their environment and/or social structure. Nevertheless, vocalizations from AFS and CFS have kept vocal similarities. As their respective distribution ranges do not overlap and individuals cannot mix for breeding, it would be interesting to further investigate the similarities in CFS and AFS vocalizations using multivariate analysis. This could complement other methods, such as genetic analysis, to understand their present taxonomic separation.

Adult CFS barks have a clear harmonic structure in contrast to those of other otariid species where the fundamental and harmonic frequency value is not measurable [24,25,80,81]. However, a clear harmonic structure for bark calls has been observed in the Subantarctic fur seal [62] which like CFS, also form large colonies that may be a constraining environment for acoustic communication. This harmonic structure which is very common in affiliative calls, may be advantageous in a noisy and confusing environment such as at high-density breeding colonies because it facilitates signal propagation and localization of the calling individual [3,23]. The production of barks in sequences may also facilitate communication in dense colonies. As predicted in the Mathematical Theory of Communication [7], using redundancy is likely to enhance the efficiency of communication and will increase the possibilities of coding information (through the duration of the sequences and the call rate for example).

### 4.1.2. Intra-species level

Comparisons of the acoustic features of FACs according to age class showed that, when getting older, pups produce longer calls and with energy more evenly distributed among harmonics. These modifications are explained by anatomic and-morphological changes over the pup's growth. Indeed, the increase in lung capacity and modifications in the size and shape of the vocal tract with age lead to changes in the filter applied to the vocalizations. In Australian fur seals, Tripovich *et al.* [27] also reported pup calls lengthening with age, but no significant change in fundamental frequency ($f0$) was found throughout the first year of life. In this study, we found some evidence for a shift in fundamental frequency over maturation for pups at PP, however, no difference was observed for CC. Although we were unable to statistically test for differences among age classes in bleating variables, a notable decrease in the proportion of pups that bleat (table 2) was found with age, suggesting a transition towards adult calls in which quavering is absent.

As bark sequences are long (average duration ranged between 4.8 and 11.8 s, table 1) and are produced by all adults and in various social contexts, they largely contribute to the natural background noise of a colony. Females use barks during agonistic interactions while males use barks mostly for territorial defence and competition with other males. It is, therefore, interesting to see if barks are acoustically different among groups of adults (sex and/or age) and assess if they can be quickly discriminated in order to improve communication among individuals and efficiency of social interactions. We first investigated differences according to sex of adults. Females produce higher-pitched barks, with shorter sequences and lower bark rate. Differences might be explained by the smaller size of the females (smaller individuals produce calls with higher fundamental frequency value, 'source-filter theory' [82]) and the use of this vocalization: conflicts among females are minor

and brief so their barks sequences do not need to be long, whereas males invest more energy in their fights. We were also interested in differences in barks between territorial males holding a harem (adult) and socially immature males (subadults). Adult males produced lower-pitched barks and have a significantly faster bark rate (i.e. shorter inter-bark duration). Differences in pitch are also likely to be linked with the body size of males [82], while the bark rate may reflect differences in the arousal state of the caller [83]. Adult males produced barks with a higher bark rate as they are more involved in aggressive interactions or related to territory defence [73,84]. In summary, the acoustic features of barks (especially the fundamental frequency value, the duration of a sequence and its bark rate) vary according to an adult's sex and social role. These characteristics among groups are undoubtedly very useful for individuals to filter the numerous signals emitted in the colony and to respond appropriately to the situation, especially when it deals with agonistic interactions.

## 4.2. Acoustic partitioning

How animals manage to communicate with conspecifics via acoustic signals in dense and highly diverse communities has long been a topic of interest. To reduce interference and to improve communication in complex acoustic environments animals can produce vocalizations either at different periods of the day (temporal partitioning) or with different acoustic characteristics (in both time and frequency domain) decreasing the probability of overlapping. Such acoustic partitioning has already been observed in animal communities e.g. insects, amphibians and birds [33,34,85–87]. A parallel can be drawn between a multi-species assemblage and a CFS breeding colony: many individuals vocalize simultaneously in very different social contexts (e.g. territoriality, competition, the attraction of a mate, conflicts and mother-young interactions). In CFS breeding colonies, adults of both sexes and pups are constantly exposed to affiliative, agonistic and mating calls produced with various grading and by emitters of different social roles and with different intents. In such an environment where the temporal overlap and the confusion risk are high, the ability for an individual to discriminate among different call types is essential to respond appropriately and is, therefore, crucial for its survival.

This study is to our knowledge the first investigation of the acoustic niche hypothesis at the intra-species level. We showed that frequency partitioning can occur among call types in the vocal repertoire of a species. From our observations, it seems that there is no temporal partitioning because all social interactions occur throughout the day and night. However, in CFS, each call type has distinguishable acoustic features and occupies a distinct niche. Particularly, short and low-pitched agonistic calls clearly differ from long and high-pitched affiliative ones. Such duration/frequency partitioning of the acoustic space improves signal transmission by facilitating its reception by the receiver. In a group, the more members interact, the more they benefit from recognizing the call type a sender is emitting as it gives crucial information on the type of social interaction (i.e. agonistic or affiliative). This acoustic partitioning is likely to constitute an advantage for group-living and colonial species for which group size is a major constraint for recognition [8,88,89].

Here our acoustic analyses showed that CFS vocalizations contain information that might allow class-level recognition [90]. This first level of recognition—referring to the ability of receivers to sort senders into categories [90]—could be then complemented by 'true' individual recognition if acoustic features of calls are individualized enough to allocate a unique identity to the sender. Further investigations on individual vocal stereotypy of CFS vocalizations and their role in individual recognition (between mother and pup or among territorial males) are thus needed.

## 4.3. Micro-geographical variations in vocalizations

Geographical variations in acoustic signals have been previously reported in pinnipeds with a special focus on phocid species [43,46,91–95] and fewer investigations for otariid species (Australian sea lion (*N. cinerea*) [48,49] and South American sea lion [50]). Such variations among colonies can have multiple origins and explanations: geographical isolation, anatomical differences between individuals, strong site fidelity, differences in acoustic propagation properties. Although the vocal repertoire was broadly similar across sites, we identified geographical variations in the acoustic features of CFS calls. In all cases, call types from CC were higher pitched compared with the same types at PP (*f0* and/or energy spectrum more widely distributed among higher frequencies). Two non-mutually exclusive hypotheses may explain these results. Firstly, differences in population density, food availability and/or genetic variations may induce differences in body size between PP and CC. Based on the 'source-filter theory' [82], smaller individuals at CC would produce calls with higher frequencies

compared with bigger seals at PP. Secondly, due to a higher population and a higher wave noise exposure (noise by breaking waves on rocks) at CC, the ambient noise level might be higher and more concentrated on low frequencies (30–500 Hz [96]). In response to these disturbances, seals might have shifted their vocalizations to higher frequency values to avoid noise overlap through an adaptive process ('acoustic adaptation hypothesis' [97]). Further investigations are needed to assess body size differences between colonies, and ambient noise level in both colonies. In the case of an environmental explanation (colony size or abiotic noise), these acoustic modifications suggest some plasticity in vocal production. Further comparisons at regional scale (macro-geographical variations) with more distant colonies (from Southern Namibia or South Africa) should clarify the geographical patterns of vocal production in this species and may provide an interesting monitoring and conservation tool.

## 5. Conclusion

This study provides the first description of the vocal repertoire of the CFS. CFS produce a limited number of call types common to other fur seal species. Their harmonic structure and their clear differences in acoustic features among call types are advantageous for signal propagation and detection through the large colonies. Variations between age and sex classes in calls facilitate discrimination among animals of different social classes and enhance social interactions in a noisy and confusing environment. The vocal characteristics of the different call types composing the vocal repertoire of CFS, therefore, are particularly well adapted to the extreme colony density of the species. Such knowledge on vocal exchanges among individuals is fundamental for a global understanding of the social system of CFSs and could be used as a basis for monitoring and management purposes.

Ethics. This present study complies with the European Union Directive on the Protection of Animals Used for Scientific Purposes (EU Directive 2010/63/EU) and with current Namibian laws. Fieldwork was permitted by the Namibian Ministry of Fisheries and Marine Resources (MFMR).

Data accessibility. Data is available at the Dryad Digital Repository: 'Cape fur seal vocal repertoire—acoustic parameters' https://doi.org/10.5061/dryad.6hdr7sqzr.

Authors' contributions. I.C. designed the study. I.C., T.G. and S.H.E. organized the fieldwork logistics. M.M. and I.C. collected the data. M.M. analysed the data. M.M. and I.C. drafted the manuscript, and all authors revised the manuscript.

Competing interests. We declare we have no competing interests.

Funding. The research was supported by CNRS and the Sea Search—Namibian Dolphin Project. The project was partially funded by the MITI-CNRS (Défi—Adaptation du vivant à son environnement 2020). M.M. is funded by a PhD scholarship from the French Ministry of Higher Education and Research. T.G. was awarded a post-doctoral fellowship from the University of Stellenbosch.

Acknowledgements. We acknowledge the Namibian Ministry of Fisheries and Marine Resources and the Namibian Chamber of Environment for their support of this research. We thank Dorothy Fourie for her assistance in the field. Special thanks to Naude Dreyer, Kevin van Schalkwyk, Craig Gibson and Jean-Paul Roux for their knowledge on the species, logistical support in the field and their enthusiasm for the project. Thank you to two anonymous reviewers whose comments substantially improved the manuscript.

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
