## [Peer Review File · Royal Society Open Science]

Review History

RSOS-202241.R0 (Original submission)

Review form: Reviewer 1

Is the manuscript scientifically sound in its present form?

Yes

Are the interpretations and conclusions justified by the results?

No

Is the language acceptable?

No

Do you have any ethical concerns with this paper?

No

Have you any concerns about statistical analyses in this paper?

No

Recommendation?

Major revision is needed (please make suggestions in comments)

Comments to the Author(s)

This manuscript describes the vocal repertoire of the Cape fur seal, which has not previously been reported. Results suggest that this species produces calls that are generally similar to those of other otariid species. Comparisons of recorded vocalizations across call type, age and sex class, and colony demonstrated that certain acoustic features were distinguishable across categories, which may enable discrimination between these groups. This work is important in terms of understanding sound production and communication in a highly colonial marine mammal.

The paper is certainly quite interesting. The manuscript presents valuable data and the recording and analysis methods are generally sound. However, in certain places explanations and corrections are needed (as noted below), and in some cases additional methodological details or clarifications should be provided. The authors should take care not to overextend the interpretations of their results, specifically with regard to the use of these calls for individual or group recognition.

This paper would be improved by careful and thorough editing. I have made note of a few of the typos and grammatical issues, but not nearly all of them. In addition to such errors, many of the sentences are quite long and convoluted, which makes it difficult for the reader to follow. This is particularly true in the Introduction and Discussion sections, which could also be simplified and streamlined for clarity.

Additional, specific feedback is provided below:

Summary

P 3, L 38-40. This is a strong statement to make for all birds and mammals, as it is certainly not true for all species. It would be more accurate to state that "The acoustic channel is an important modality used by birds and mammals..."

P 3, L 48-50. The authors should be careful (here and throughout the ms) to differentiate between calls having the potential to be used for discrimination (among individuals, age or sex classes, etc.) and calls actually being used in that way. The latter would need to be documented through targeted behavioral studies with this species.

P 3, L 51-53. This assertion depends on how "distinct acoustic niches" are defined – just because there are some significant differences in spectral and temporal patterns across call types, this doesn't necessarily mean that the calls occupy distinct acoustic space. Clear evidence needs to be provided (in the main body of the ms) to support this claim.

Introduction

P 4, L 7. Suggest "fundamental" instead of "primordial," here and elsewhere.

P 4, L 15. Communication can involve other sensory modalities as well. To be complete, suggest including all of these here, or adding "e.g.," inside the parentheses.

P 4, L 18-20. These general statements about acoustic communication are sometimes true, but not always. It is not always true, for example, that acoustic signals propagate over long distances or that they can be easily localized (these depend on the temporal and spectral characteristics of the signal, the environment they're produced in, and the auditory capabilities of any listeners).

P 4, L 20-21. Again, whether sound is the most efficient sensory cue for communication really depends on the environment, the signal of interest, and the sensory capabilities of the intended receiver(s). It is not always true for all species, distances (between signaler and receiver),

conditions, etc. This is discussed later in the paragraph in terms of the factors limiting the effectiveness of vocal exchanges. For accuracy, these statements should be revised.

P 4, L 42. What is meant here by encoding-decoding mechanisms? Is this referring to 1) mechanisms of sound production, and 2) the sensory and cognitive processing capabilities of the relevant species? Something else? Suggest explaining more thoroughly or re-phrasing so the reader can follow more easily if unfamiliar with this specific terminology.

P 4, L 45-47. Suggest defining taxa that the reader might not be familiar with (e.g., cetacean, phocid) or being more consistent with naming conventions across taxa (e.g., whales or seals could be used to be more parallel).

P 4, L 47-53; P 5, L 8-18; P 5, L 33-38; P 5-6, L 54-3. Some examples of when writing is a bit dense and difficult to follow.

P 4, L 54-56. Please provide the scientific name of the Cape fur seal. Also suggest defining Otariid, for example with "Otariid (eared seal)," here.

P 4, L 56-58. Suggest "...because of the extreme density of colonies and their social structure during the breeding season, which require..."

P 5, L 3-8. Suggest removing these details about Cape Cross, since they're also included in the Methods section.

P 5, L 19. Suggest defining pinnipeds here for readers who are unfamiliar.

P 5, L 28-29. This should read "...initial investigations on a larger scale and explored..."

P 5, L 31-32. Suggest something like "...we aimed to assess whether the signal contained unique acoustic features which could potentially be used to convey information about..."

P 5, L 33-38. What is the basis for this hypothesis? Has this been shown in other species? Are there relevant studies the authors could refer to here to support this hypothesis?

P 5, L 41. Suggest deleting "in both study sites" since the study sites have not been introduced yet.

P 5, L 44-47. Suggest framing this more as a challenge or constraint these animals face, rather than something that they certainly do.

P 6, L 19. It would be helpful to define micro- and macro-geographic variations (here or elsewhere) since these are important concepts in this paper.

Materials and Methods

P 6, L 29. In this section, the different sex and age classes of CFS recorded at each colony should be clearly described.

P 6, L 34-35. What is the age range of subadult males? Less than 9 yo and greater than xxx?? Was this determined through observation of any physical characteristics, or based only on whether or not the male formed a harem? What if a male was sexually and socially mature, but did not establish a territory or form a harem this particular year? Please provide more details.

P 7, L 8-14. Since directional microphones were used, were all calls recorded on axis? If not, how might this have affected the measured acoustic features of these calls? Please provide more details here.

P 7, L 12-13. How accurately can the sex of pups be determined visually, at a distance? Was age class always determined, or only for males? Were female subadults not recorded, or were they grouped with adults, and why?

P 7, L 16+. Was this age estimation of pups based on other studies -- can references be provided? And again, how were older animals categorized? More details are needed.

P 7, L 30-31. Was the recording distance for each call noted or considered in the analysis? Propagation through the colony could have modified the frequency characteristics of the recorded vocals, which would have an important impact on the results of this study.

P 7, L 28-29. Did the experimenters typically observe any behavioral disturbance upon their arrival? Was this 15-minute acclimation period after the experimenter(s) arrived at the colony, or after behavior returned to baseline?

P 7, L 42-43. Please provide frequency resolution with these settings and sampling rate (i.e., 3-dB filter bandwidth).

P 7, L 42-43. How were “good quality” calls defined? Was it just based on SNR? Were incomplete (partially recorded) calls considered in this analysis.

P 7, L 44-45. Can “low background noise” be quantified? Was there an SNR cutoff? How much did ambient noise conditions vary day to day, and did they vary significantly across colonies? Providing information about the noise background at each colony (or at least more quantitative information about acceptable SNRs) is necessary to ensure that this did not have any effect on the results.

P 7, L 47-48. Why was 100 Hz chosen for the high-pass filter cutoff? According to Table 1, multiple call types have fundamental frequencies close to 100 Hz. In fact, growls produced by females at Pelican Point had a mean fundamental frequency of 90 Hz. This seems problematic for this dataset, and should be addressed or corrected.

P 7, L 49. Were calls categorized into types based on common perceptual features, or based on context? Please clarify. The way this is written, it sounds like context was the main driver for call classification.

P 7, L 60. There is a period missing at the end of the last full sentence on this page.

P 8, L 6-7. Again, frequency resolution (3-dB bandwidth) with these settings would be useful to provide here.

P 8, L 15-16. This is a little unclear. These additional variables were specific to a given call type AND were not always measurable for all calls of that type? Or should that be “or” there?

P 8, L 23-34. Can bleating/quavering be defined here or elsewhere? A reference is provided, but this would still be helpful.

P 8, L 28-29. It's confusing here, and elsewhere throughout the Methods, that the call types are mentioned/alluded to but not defined. Suggest defining the five main call types in the Methods if they're going to be discussed at all.

P 8, L 30. Were soft versus aggressive growls context specific, or was the only difference the amplitude of the calls? It's a bit misleading that one name refers to (relative) amplitude and the other refers to context, and that that two are mutually exclusive. Were “soft growls” ever produced aggressively? How loud did the call have to be to be considered soft versus aggressive? Please provide more explanation.

P 8, L 52-53. Can a reference be provided for the Random Forest algorithm?

P 9, L 36. FAC has not been defined yet.

P 9, L 56. Up until now, the methods have seemed to focus on pups. It was not completely clear before this point whether adults were actually recorded. This needs to be specified earlier (which age and sex classes were recorded and, again, how they were defined).

P 9, L 58. Again, please make sure that distinctions between age classes (if any) are clearly described for both males and females.

P 10, L 11-13. Adults and subadults in these comparisons are presumably all males? Please clarify.

P 10, L 26-27. These calls types have not been identified or described yet. These abbreviations have not been defined, so should not be used here.

Results

P 10, L 49. Do the males and females listed here represent all age classes (aside from pups)?

P 10, L 50. It would be useful to provide the relative proportions of each call type in the Results (either in the main text or in Table 1).

P 11, L 9-10. What are these values? Previously the authors stated that mean and stdev were used, so why is a range provided here? Based on Table 1, I believe these are the means for the two colonies, and the standard deviations for the two colonies. This must be explicitly stated, especially since this is not standard notation. Furthermore, the data from the two colonies are not

always presented in the same order, so this should be noted as well (data are provided from low to high instead).

P 11, L 24-25. How did sample size compare for the two call types? Could this have affected their relative standard deviations?

P 11, L 3. Why were some of the prediction plots included as supplements and some as part of the main text? Could these be combined into a multi-panel figure potentially?

P 11, L 49. The pup age classes were described in the Methods, but I don't believe they were explicitly assigned numbers (age classes 1 through 3). Please confirm and define in the Methods if necessary.

P 11, L 49-50. These scatterplots seem to show quite a bit of overlap between all three age classes, and it's not obvious that age classes 1 and 3 are "quite distinct" acoustically.

P 12, L 31-32. Should read, "...accuracy of prediction values were 83% (CC) and 78% (PP)..." The authors should check tenses and number agreement (between nouns and verbs) throughout the manuscript for accuracy. There are a lot of issues with this.

P 12, L 47-48. To clarify, were subadult versus adult females not separated by age class?

Presumably not, but this is not explicitly stated anywhere. Also, in some places "adult females" is specified. It is unclear whether subadult and adult females were grouped together, or if subadult females were not considered. These details need to be included, clearly, in the Methods.

P 13, L 43-47. Suggest revising and splitting into two sentences.

P 13, L 50. Should this be "Micro-geographic variation in vocalizations"

P 13, L 52-55. Why were these particular call type/sex/age class combinations chosen for comparison? Are these adult females or all females?

Discussion

P 14, L 18-19. Suggest "...overall structure and biological function are apparently similar..."

P 14, L 25. Fond should be "find" in this sentence.

P 14, L 29-30. What was the sex ratio at each colony? Data should be provided in the Results to support this claim.

P 14, L 48-49. Suggest replacing "genders" with "sexes."

P 14, L 52. Can the estimated divergence time be provided (X mya)?

P 14, L 52. Or, were these calls added to the AFS repertoire later? Unless we know something about vocal behavior in their most recent common ancestor, this can't really be determined here. This statement should be revised.

P 16, L 24-27. There are multiple colons in this sentence, suggest re-organizing.

P 16, L 41-49. While the call types have some recognizable (and unique) acoustic features, it is not clear that this constitutes a "distinct niche" for each call type. There is still overlap between types, and across age and sex classes. Are the authors referring to individual recognition here (L 46)?

Because these data do not speak to that. Importantly, just because there are differences in acoustic features across call types, this does not necessarily mean that receivers are utilizing this information for individual, sex, or age class recognition. That would need to be further investigated throughout behavioral studies (e.g., playback experiments). The authors should be careful not to overextend the interpretation of their results.

P 16, L 51. Suggest "...CFS vocalizations contain information..." instead of "transmit information." While there might be differences across calls that provide information, it is unclear the extent to which that information is actually used by receivers in this species.

P 17, L 35. This is certainly a possibility, but are the individuals at Cape Cross actually smaller, or is there reason to suspect that this may be the case?

P 17, L 23-26. Are ambient noise levels relatively higher at Cape Cross, or concentrated at low frequencies? This is another reason to discuss ambient noise levels in the Methods or Results.

P 17, L 48-49. Not really. This would just facilitate discrimination among age and sex classes (not individuals).

P 17, L 50-51. Is it really true that the vocal repertoire of CFS is “particularly adapted to the extreme colony density” – how does the repertoire compare to a related species with less dense colonies?

Tables

Table 1 caption. For the significance code, 5 significance values are provided but only 4 labels are given. Please check for accuracy (here and elsewhere). Also suggest a more explicit explanatory sentence. What does NA refer to in this context? Suggest replacing “brackets” with “parentheses” in caption.

Table 1. Are these females (for barks and growls) of any age?

Table 2. What age classes are between 4 months (pups) and adult females? The age structure used in this study needs to be more clearly described.

Figures

Figure 1. Please include spectrogram parameters in the caption. Caption for panel b: these are presumably provided in age order from left to right, but please clarify. To facilitate comparison, suggest plotting the different call types on common time and frequency axes – otherwise, note in the caption that they are plotted on different axes.

Figure 2. Are these data pooled across age and sex classes and behavioral contexts? Details such as this should be provided in the figure captions so the reader doesn’t have to refer back to the main text.

Figure 3. In the caption, suggest including the variables that contributed most to the total variation, for reference.

Figure 6. This figure is much more convincing than the other LDA scatterplots, which show a lot more overlap between call types/age or sex classes (e.g., Figure 5).

Review form: Reviewer 2

Is the manuscript scientifically sound in its present form?

Yes

Are the interpretations and conclusions justified by the results?

Yes

Is the language acceptable?

Yes

Do you have any ethical concerns with this paper?

No

Have you any concerns about statistical analyses in this paper?

No

Recommendation?

Accept with minor revision (please list in comments)

Comments to the Author(s)

RSOS-202241

Title: Vocal repertoire, micro-geographic variation and within-species acoustic partitioning in a highly colonial pinniped, the Cape fur seal.

This manuscript is interesting and provides proper information about the in air vocal repertoire of the Cape fur seal, as well as micro-geographic variation and within-species acoustic partitioning in this highly colonial pinniped species. In my opinion, parts of the description of the context, and methods could do with a little more detail or explanation, but otherwise I think the manuscript is highly valuable.

Note to the authors: Please next time use actual line numbers for an easier review.

Main comments: I found some background material to be lacking in the introduction/methods – I did not have a good sense of the two locations, the density of Cape fur seal in the areas, and I did not have a clear sense of the animals that would be in acoustic or visual range.

Please provide scientific names for each species [for example Page 4: South American fur seal [47,48], Australian fur seal [49], Subantarctic fur seal [50] and Northern fur seal [51].]

Specific recommendations are provided below:

ABSTRACT

“We described the acoustic features and social function of five IN AIR call types...”

INTRODUCTION

Page 2: Second paragraph: You should mention that your first goal focus during the breeding season.

Page 2: third paragraph : “... it is an extremely noisy environment combining a high risk of vocal and visual confusion among conspecifics.”. Visual? Do you mean “auditory” ?- A noisy environment should not affect the visual perception. Also, what do you mean by vocal confusion? Do you mean that the animal does not know which sound produce? I would suggest rephrasing this sentence.

Page 2: third paragraph: Does different group age, sex and social status are mixed/grouped in the colony?

MATERIAL AND METHODS

Page 3: 3.1 First paragraph – I would suggest moving this paragraph in the introduction.

Page 3: 3.1 Second paragraph – What is the distance (km) between sites? Do Cape fur seal stay all year round near to their colonies or can they switch from one colony to the other? How many Cape fur seal colonies do you have along the Namibian coast? I would suggest adding a figure with a map showing both locations (and potentially some other colonies) and including some pictures of each location to provide a better background to the reader.

You said that the breeding is a 4 month period, how many times did you go in the field for the recordings? Maybe provide the number of days that you spent to each location?

Page 4: 3.2 - “A maximum of 10 calls per individual were included in the analysis”. Is it a combination of call types or only one call type? (10 of the same call type or any call type?)

“Signals were categorized into different call types based on the behavioural context of production...” Please provide the call types here. Later in the paragraph and the next one, you mention barks, pup-attraction calls and growl.

Page 5: “..., measurements were thus performed on 5 barks randomly chosen ...” – Does the 5 barks are included in the maximum of 10 calls?

Page 9: “trees were built with the same sample size for each group (i.e. the size of the smallest group): n=135 for Cape Cross and n=150 for Pelican Point. Does n is the number of individuals or the number of recorded barks?

RESULTS

Page 9 – “Differences between sexes were only investigated for very young pups at Pelican Point”. Why not for older pups?

DISCUSSION

Page 11: “The role of this call remains unknown but could be related to a high level of stress or emotional state”. Speculative. Please provide more justification (why animals would be stressed...etc) or delete it.

Page 11: "Cape fur seal and Australian fur seal ... are two sub-species of *A. pusillus*"

FIGURE

Figure 1. Please provide arguments used for the spectrogram (window, overlap, fft). Also I would suggest to include some audio files corresponding to each CFS call type as supplemental material.

Figure 5: adult male and females and subadult males

Decision letter (RSOS-202241.R0)

Dear Mrs Martin

The Editors assigned to your paper RSOS-202241 "Vocal repertoire, micro-geographic variation and within-species acoustic partitioning in a highly colonial pinniped, the Cape fur seal" have now received comments from reviewers and would like you to revise the paper in accordance with the reviewer comments and any comments from the Editors. Please note this decision does not guarantee eventual acceptance.

Please submit your revised manuscript and required files (see below) no later than 21 days from today's (ie 19-Apr-2021) date. Note: the ScholarOne system will 'lock' if submission of the revision is attempted 21 or more days after the deadline. If you do not think you will be able to meet this deadline please contact the editorial office immediately.

Best regards,

on behalf of Dr Ari Friedlaender (Associate Editor) and Kevin Padian (Subject Editor)
openscience@royalsociety.org

Associate Editor Comments to Author (Dr Ari Friedlaender):

To the Authors,

Both reviewers provide excellent feedback and commentary on your submission. As a first description of the vocalizations of the species there is interest in the results, however, it appears that there are major concerns raised regarding the interpretations of the findings and whether there is justification to be able to place context to certain signals given the data that are presented. I would like the authors to carefully consider the language that is used for interpreting the different acoustic signals described in the submission and what, if any, meaning can be afforded to them. As well, both reviewers found significant and substantial need for major editorial work on sentence structure, language, and clarity. Please consider this carefully and be diligent when revising your submission to be as precise as possible with the text. I am interested to read a revised version of this work and how the reviewer comments are incorporated to more appropriately interpret the findings.

Thank you.
Ari S. Friedlaender

Reviewer comments to Author:

Reviewer: 1

Comments to the Author(s)

This manuscript describes the vocal repertoire of the Cape fur seal, which has not previously been reported. Results suggest that this species produces calls that are generally similar to those of other otariid species. Comparisons of recorded vocalizations across call type, age and sex class, and colony demonstrated that certain acoustic features were distinguishable across categories, which may enable discrimination between these groups. This work is important in terms of understanding sound production and communication in a highly colonial marine mammal.

The paper is certainly quite interesting. The manuscript presents valuable data and the recording and analysis methods are generally sound. However, in certain places explanations and corrections are needed (as noted below), and in some cases additional methodological details or clarifications should be provided. The authors should take care not to overextend the interpretations of their results, specifically with regard to the use of these calls for individual or group recognition.

This paper would be improved by careful and thorough editing. I have made note of a few of the typos and grammatical issues, but not nearly all of them. In addition to such errors, many of the sentences are quite long and convoluted, which makes it difficult for the reader to follow. This is particularly true in the Introduction and Discussion sections, which could also be simplified and streamlined for clarity.

Additional, specific feedback is provided below:

Summary

P 3, L 38-40. This is a strong statement to make for all birds and mammals, as it is certainly not true for all species. It would be more accurate to state that "The acoustic channel is an important modality used by birds and mammals..."

P 3, L 48-50. The authors should be careful (here and throughout the ms) to differentiate between calls having the potential to be used for discrimination (among individuals, age or sex classes, etc.) and calls actually being used in that way. The latter would need to be documented through targeted behavioral studies with this species.

P 3, L 51-53. This assertion depends on how “distinct acoustic niches” are defined – just because there are some significant differences in spectral and temporal patterns across call types, this doesn’t necessarily mean that the calls occupy distinct acoustic space. Clear evidence needs to be provided (in the main body of the ms) to support this claim.

Introduction

P 4, L 7. Suggest “fundamental” instead of “primordial,” here and elsewhere.

P 4, L 15. Communication can involve other sensory modalities as well. To be complete, suggest including all of these here, or adding “e.g.,” inside the parentheses.

P 4, L 18-20. These general statements about acoustic communication are sometimes true, but not always. It is not always true, for example, that acoustic signals propagate over long distances or that they can be easily localized (these depend on the temporal and spectral characteristics of the signal, the environment they’re produced in, and the auditory capabilities of any listeners).

P 4, L 20-21. Again, whether sound is the most efficient sensory cue for communication really depends on the environment, the signal of interest, and the sensory capabilities of the intended receiver(s). It is not always true for all species, distances (between signaler and receiver), conditions, etc. This is discussed later in the paragraph in terms of the factors limiting the effectiveness of vocal exchanges. For accuracy, these statements should be revised.

P 4, L 42. What is meant here by encoding-decoding mechanisms? Is this referring to 1) mechanisms of sound production, and 2) the sensory and cognitive processing capabilities of the relevant species? Something else? Suggest explaining more thoroughly or re-phrasing so the reader can follow more easily if unfamiliar with this specific terminology.

P 4, L 45-47. Suggest defining taxa that the reader might not be familiar with (e.g., cetacean, phocid) or being more consistent with naming conventions across taxa (e.g., whales or seals could be used to be more parallel).

P 4, L 47-53; P 5, L 8-18; P 5, L 33-38; P 5-6, L 54-3. Some examples of when writing is a bit dense and difficult to follow.

P 4, L 54-56. Please provide the scientific name of the Cape fur seal. Also suggest defining Otariid, for example with “Otariid (eared seal),” here.

P 4, L 56-58. Suggest “...because of the extreme density of colonies and their social structure during the breeding season, which require...”

P 5, L 3-8. Suggest removing these details about Cape Cross, since they’re also included in the Methods section.

P 5, L 19. Suggest defining pinnipeds here for readers who are unfamiliar.

P 5, L 28-29. This should read “...initial investigations on a larger scale and explored...”

P 5, L 31-32. Suggest something like “...we aimed to assess whether the signal contained unique acoustic features which could potentially be used to convey information about...”

P 5, L 33-38. What is the basis for this hypothesis? Has this been shown in other species? Are there relevant studies the authors could refer to here to support this hypothesis?

P 5, L 41. Suggest deleting “in both study sites” since the study sites have not been introduced yet.

P 5, L 44-47. Suggest framing this more as a challenge or constraint these animals face, rather than something that they certainly do.

P 6, L 19. It would be helpful to define micro- and macro-geographic variations (here or elsewhere) since these are important concepts in this paper.

Materials and Methods

P 6, L 29. In this section, the different sex and age classes of CFS recorded at each colony should be clearly described.

P 6, L 34-35. What is the age range of subadult males? Less than 9 yo and greater than xxx?? Was this determined through observation of any physical characteristics, or based only on whether or not the male formed a harem? What if a male was sexually and socially mature, but did not establish a territory or form a harem this particular year? Please provide more details.

P 7, L 8-14. Since directional microphones were used, were all calls recorded on axis? If not, how might this have affected the measured acoustic features of these calls? Please provide more details here.

P 7, L 12-13. How accurately can the sex of pups be determined visually, at a distance? Was age class always determined, or only for males? Were female subadults not recorded, or were they grouped with adults, and why?

P 7, L 16+. Was this age estimation of pups based on other studies -- can references be provided? And again, how were older animals categorized? More details are needed.

P 7, L 30-31. Was the recording distance for each call noted or considered in the analysis? Propagation through the colony could have modified the frequency characteristics of the recorded vocals, which would have an important impact on the results of this study.

P 7, L 28-29. Did the experimenters typically observe any behavioral disturbance upon their arrival? Was this 15-minute acclimation period after the experimenter(s) arrived at the colony, or after behavior returned to baseline?

P 7, L 42-43. Please provide frequency resolution with these settings and sampling rate (i.e., 3-dB filter bandwidth).

P 7, L 42-43. How were "good quality" calls defined? Was it just based on SNR? Were incomplete (partially recorded) calls considered in this analysis.

P 7, L 44-45. Can "low background noise" be quantified? Was there an SNR cutoff? How much did ambient noise conditions vary day to day, and did they vary significantly across colonies? Providing information about the noise background at each colony (or at least more quantitative information about acceptable SNRs) is necessary to ensure that this did not have any effect on the results.

P 7, L 47-48. Why was 100 Hz chosen for the high-pass filter cutoff? According to Table 1, multiple call types have fundamental frequencies close to 100 Hz. In fact, growls produced by females at Pelican Point had a mean fundamental frequency of 90 Hz. This seems problematic for this dataset, and should be addressed or corrected.

P 7, L 49. Were calls categorized into types based on common perceptual features, or based on context? Please clarify. The way this is written, it sounds like context was the main driver for call classification.

P 7, L 60. There is a period missing at the end of the last full sentence on this page.

P 8, L 6-7. Again, frequency resolution (3-dB bandwidth) with these settings would be useful to provide here.

P 8, L 15-16. This is a little unclear. These additional variables were specific to a given call type AND were not always measurable for all calls of that type? Or should that be "or" there?

P 8, L 23-34. Can bleating/quavering be defined here or elsewhere? A reference is provided, but this would still be helpful.

P 8, L 28-29. It's confusing here, and elsewhere throughout the Methods, that the call types are mentioned/alluded to but not defined. Suggest defining the five main call types in the Methods if they're going to be discussed at all.

P 8, L 30. Were soft versus aggressive growls context specific, or was the only difference the amplitude of the calls? It's a bit misleading that one name refers to (relative) amplitude and the other refers to context, and that that two are mutually exclusive. Were "soft growls" ever produced aggressively? How loud did the call have to be to be considered soft versus aggressive? Please provide more explanation.

P 8, L 52-53. Can a reference be provided for the Random Forest algorithm?

P 9, L 36. FAC has not been defined yet.

P 9, L 56. Up until now, the methods have seemed to focus on pups. It was not completely clear before this point whether adults were actually recorded. This needs to be specified earlier (which age and sex classes were recorded and, again, how they were defined).

P 9, L 58. Again, please make sure that distinctions between age classes (if any) are clearly described for both males and females.

P 10, L 11-13. Adults and subadults in these comparisons are presumably all males? Please clarify.

P 10, L 26-27. These calls types have not been identified or described yet. These abbreviations have not been defined, so should not be used here.

Results

P 10, L 49. Do the males and females listed here represent all age classes (aside from pups)?

P 10, L 50. It would be useful to provide the relative proportions of each call type in the Results (either in the main text or in Table 1).

P 11, L 9-10. What are these values? Previously the authors stated that mean and stdev were used, so why is a range provided here? Based on Table 1, I believe these are the means for the two colonies, and the standard deviations for the two colonies. This must be explicitly stated, especially since this is not standard notation. Furthermore, the data from the two colonies are not always presented in the same order, so this should be noted as well (data are provided from low to high instead).

P 11, L 24-25. How did sample size compare for the two call types? Could this have affected their relative standard deviations?

P 11, L 3. Why were some of the prediction plots included as supplements and some as part of the main text? Could these be combined into a multi-panel figure potentially?

P 11, L 49. The pup age classes were described in the Methods, but I don't believe they were explicitly assigned numbers (age classes 1 through 3). Please confirm and define in the Methods if necessary.

P 11, L 49-50. These scatterplots seem to show quite a bit of overlap between all three age classes, and it's not obvious that age classes 1 and 3 are "quite distinct" acoustically.

P 12, L 31-32. Should read, "...accuracy of prediction values were 83% (CC) and 78% (PP)..." The authors should check tenses and number agreement (between nouns and verbs) throughout the manuscript for accuracy. There are a lot of issues with this.

P 12, L 47-48. To clarify, were subadult versus adult females not separated by age class?

Presumably not, but this is not explicitly stated anywhere. Also, in some places "adult females" is specified. It is unclear whether subadult and adult females were grouped together, or if subadult females were not considered. These details need to be included, clearly, in the Methods.

P 13, L 43-47. Suggest revising and splitting into two sentences.

P 13, L 50. Should this be "Micro-geographic variation in vocalizations"

P 13, L 52-55. Why were these particular call type/sex/age class combinations chosen for comparison? Are these adult females or all females?

Discussion

P 14, L 18-19. Suggest "...overall structure and biological function are apparently similar..."

P 14, L 25. Fond should be "find" in this sentence.

P 14, L 29-30. What was the sex ratio at each colony? Data should be provided in the Results to support this claim.

P 14, L 48-49. Suggest replacing "genders" with "sexes."

P 14, L 52. Can the estimated divergence time be provided (X mya)?

P 14, L 52. Or, were these calls added to the AFS repertoire later? Unless we know something about vocal behavior in their most recent common ancestor, this can't really be determined here. This statement should be revised.

P 16, L 24-27. There are multiple colons in this sentence, suggest re-organizing.

P 16, L 41-49. While the call types have some recognizable (and unique) acoustic features, it is not clear that this constitutes a “distinct niche” for each call type. There is still overlap between types, and across age and sex classes. Are the authors referring to individual recognition here (L 46)?

Because these data do not speak to that. Importantly, just because there are differences in acoustic features across call types, this does not necessarily mean that receivers are utilizing this information for individual, sex, or age class recognition. That would need to be further investigated throughout behavioral studies (e.g., playback experiments). The authors should be careful not to overextend the interpretation of their results.

P 16, L 51. Suggest “...CFS vocalizations contain information...” instead of “transmit information.” While there might be differences across calls that provide information, it is unclear the extent to which that information is actually used by receivers in this species.

P 17, L 35. This is certainly a possibility, but are the individuals at Cape Cross actually smaller, or is there reason to suspect that this may be the case?

P 17, L 23-26. Are ambient noise levels relatively higher at Cape Cross, or concentrated at low frequencies? This is another reason to discuss ambient noise levels in the Methods or Results.

P 17, L 48-49. Not really. This would just facilitate discrimination among age and sex classes (not individuals).

P 17, L 50-51. Is it really true that the vocal repertoire of CFS is “particularly adapted to the extreme colony density” – how does the repertoire compare to a related species with less dense colonies?

Tables

Table 1 caption. For the significance code, 5 significance values are provided but only 4 labels are given. Please check for accuracy (here and elsewhere). Also suggest a more explicit explanatory sentence. What does NA refer to in this context? Suggest replacing “brackets” with “parentheses” in caption.

Table 1. Are these females (for barks and growls) of any age?

Table 2. What age classes are between 4 months (pups) and adult females? The age structure used in this study needs to be more clearly described.

Figures

Figure 1. Please include spectrogram parameters in the caption. Caption for panel b: these are presumably provided in age order from left to right, but please clarify. To facilitate comparison, suggest plotting the different call types on common time and frequency axes – otherwise, note in the caption that they are plotted on different axes.

Figure 2. Are these data pooled across age and sex classes and behavioral contexts? Details such as this should be provided in the figure captions so the reader doesn’t have to refer back to the main text.

Figure 3. In the caption, suggest including the variables that contributed most to the total variation, for reference.

Figure 6. This figure is much more convincing than the other LDA scatterplots, which show a lot more overlap between call types/age or sex classes (e.g., Figure 5).

Reviewer: 2

Comments to the Author(s)

RSOS-202241

Title: Vocal repertoire, micro-geographic variation and within-species acoustic partitioning in a highly colonial pinniped, the Cape fur seal.

This manuscript is interesting and provides proper information about the in air vocal repertoire of the Cape fur seal, as well as micro-geographic variation and within-species acoustic partitioning in this highly colonial pinniped species. In my opinion, parts of the description of the context, and methods could do with a little more detail or explanation, but otherwise I think the manuscript is highly valuable.

Note to the authors: Please next time use actual line numbers for an easier review.

Main comments: I found some background material to be lacking in the introduction/methods – I did not have a good sense of the two locations, the density of Cape fur seal in the areas, and I did not have a clear sense of the animals that would be in acoustic or visual range.

Please provide scientific names for each species [for example Page 4: South American fur seal [47,48], Australian fur seal [49], Subantarctic fur seal [50] and Northern fur seal [51].]

Specific recommendations are provided below:

ABSTRACT

“We described the acoustic features and social function of five IN AIR call types...”

INTRODUCTION

Page 2: Second paragraph: You should mention that your first goal focus during the breeding season.

Page 2: third paragraph : “... it is an extremely noisy environment combining a high risk of vocal and visual confusion among conspecifics.”. Visual? Do you mean “auditory” ?– A noisy environment should not affect the visual perception. Also, what do you mean by vocal confusion? Do you mean that the animal does not know which sound produce? I would suggest rephrasing this sentence.

Page 2: third paragraph: Does different group age, sex and social status are mixed/grouped in the colony?

MATERIAL AND METHODS

Page 3: 3.1 First paragraph – I would suggest moving this paragraph in the introduction.

Page 3: 3.1 Second paragraph – What is the distance (km) between sites? Do Cape fur seal stay all year round near to their colonies or can they switch from one colony to the other? How many Cape fur seal colonies do you have along the Namibian coast? I would suggest adding a figure with a map showing both locations (and potentially some other colonies) and including some pictures of each location to provide a better background to the reader.

You said that the breeding is a 4 month period, how many times did you go in the field for the recordings? Maybe provide the number of days that you spent to each location?

Page 4: 3.2 - “A maximum of 10 calls per individual were included in the analysis”. Is it a combination of call types or only one call type? (10 of the same call type or any call type?)

“Signals were categorized into different call types based on the behavioural context of production...” Please provide the call types here. Later in the paragraph and the next one, you mention barks, pup-attraction calls and growl.

Page 5: “..., measurements were thus performed on 5 barks randomly chosen ...” – Does the 5 barks are included in the maximum of 10 calls?

Page 9: “trees were built with the same sample size for each group (i.e. the size of the smallest group): n=135 for Cape Cross and n=150 for Pelican Point. Does n is the number of individuals or the number of recorded barks?

RESULTS

Page 9 – “Differences between sexes were only investigated for very young pups at Pelican Point”. Why not for older pups?

DISCUSSION

Page 11: “The role of this call remains unknown but could be related to a high level of stress or emotional state”. Speculative. Please provide more justification (why animals would be stressed...etc) or delete it.

Page 11: “Cape fur seal and Australian fur seal ... are two sub-species of *A. pusillus*”

FIGURE

Figure 1. Please provide arguments used for the spectrogram (window, overlap, fft). Also I would suggest to include some audio files corresponding to each CFS call type as supplemental material. Figure 5: adult male and females and subadult males

===PREPARING YOUR MANUSCRIPT===

Your revised paper should include the changes requested by the referees and Editors of your manuscript. You should provide two versions of this manuscript and both versions must be provided in an editable format:
 one version identifying all the changes that have been made (for instance, in coloured highlight, in bold text, or tracked changes);
 a 'clean' version of the new manuscript that incorporates the changes made, but does not highlight them. This version will be used for typesetting if your manuscript is accepted.

===PREPARING YOUR REVISION IN SCHOLARONE===

<https://royalsociety.org/journals/authors/author-guidelines/#supplementary-material> to include a suitable title and informative caption. An example of appropriate titling and captioning may be found at [https://figshare.com/articles/Table_S2_from_Is_there_a_trade-off_between_peak_performance_and_performance_breadth_across_temperatures_for_aerobic_sc ope_in_teleost_fishes_/3843624](https://figshare.com/articles/Table_S2_from_Is_there_a_trade-off_between_peak_performance_and_performance_breadth_across_temperatures_for_aerobic_scope_in_teleost_fishes_/3843624).

Author's Response to Decision Letter for (RSOS-202241.R0)

See Appendix A.

RSOS-202241.R1 (Revision)

Review form: Reviewer 1

Is the manuscript scientifically sound in its present form?

Yes

Are the interpretations and conclusions justified by the results?

Yes

Is the language acceptable?

Yes

Do you have any ethical concerns with this paper?

No

Have you any concerns about statistical analyses in this paper?

No

Recommendation?

Accept as is

Comments to the Author(s)

This manuscript has been greatly improved through editing, and the authors have responded sufficiently to address all of my prior concerns. This is an interesting paper and merits publication in this journal.

A few additional, minor edits are included below (note that line numbers correspond to the version with track changes):

Introduction

This section lays out background info and study aims well.

L 21. Should read "Communication is rarely a uni-modal..."

L 37. Should read "...to a receiver through a signal in which..."

L 64. Should read "Females also engage in agonistic interactions..."

L 87-88. Suggest putting all text from i.e. to end of sentence in parentheses.

L 101-102. Suggest, "This could allow individuals to quickly identify the social role of a caller, (i.e. the age, sex, or social position of a calling conspecific).

L 115. Should be "...both micro- and macro-geographic variation have been found..."

L 117. Should be "...pups' contact calls of South American sea lions..."

Materials and Methods

The additions in the Animals and recording procedure section are helpful!

L 123-127. It's a little confusing that two slightly different age ranges for social maturity are provided in this section - suggest removing one.

L 135. Closed parenthesis missing.

L 185. Should read "...previous knowledge of other..."

L 196. "500 Hz for adults' calls..."

L 250. Suggest "...number of good-quality, complete calls in the smallest..."

L 277. Should this be "...considering sequences with barks..."

Results

L 300. electronic supplementary material

L 411. Micro-geographic variations in vocalizations

L 417. Could the potentially louder low-frequency noise at Cape Cross have masked any low-frequency call components?

Discussion

L 535. Missing closed parenthesis.

Conclusion.

L 562. Suggest "...therefore are particularly well adapted to the extreme..."

Figures

Figure 4. Y-axis should be indicator of precision

Electronic supplementary Audio S1. Suggest verbally annotating this audio file to make it easier for the reader to identify call types and match to Figure 1. This is great to include!

Electronic supplementary Figure S2. Update labels of age classes to reflect ms terminology (these are no longer used).

Review form: Reviewer 2

Is the manuscript scientifically sound in its present form?

Yes

Are the interpretations and conclusions justified by the results?

Yes

Is the language acceptable?

Yes

Do you have any ethical concerns with this paper?

No

Have you any concerns about statistical analyses in this paper?

No

Recommendation?

Accept as is

Comments to the Author(s)

The authors have adequately revised the manuscript and addressed the comments of both reviewers. One last question: can you add the number of recording days (effort) at both locations in the manuscript? I saw your reply but cannot find the information in the manuscript.

Decision letter (RSOS-202241.R1)

Dear Mrs Martin,

It is a pleasure to accept your manuscript entitled "Vocal repertoire, micro-geographic variation and within-species acoustic partitioning in a highly colonial pinniped, the Cape fur seal" in its current form for publication in Royal Society Open Science. The comments of the reviewer(s) who reviewed your manuscript are included at the foot of this letter.

on behalf of Dr Ari Friedlaender (Associate Editor) and Kevin Padian (Subject Editor)
openscience@royalsociety.org

Reviewer comments to Author:

Reviewer: 1

Comments to the Author(s)

This manuscript has been greatly improved through editing, and the authors have responded sufficiently to address all of my prior concerns. This is an interesting paper and merits publication in this journal.

A few additional, minor edits are included below (note that line numbers correspond to the version with track changes):

Introduction

This section lays out background info and study aims well.

L 21. Should read "Communication is rarely a uni-modal..."

- L 37. Should read "...to a receiver through a signal in which..."
- L 64. Should read "Females also engage in agonistic interactions..."
- L 87-88. Suggest putting all text from i.e. to end of sentence in parentheses.
- L 101-102. Suggest, "This could allow individuals to quickly identify the social role of a caller, (i.e. the age, sex, or social position of a calling conspecific).
- L 115. Should be "...both micro- and macro-geographic variation have been found..."
- L 117. Should be "...pups' contact calls of South American sea lions..."

Materials and Methods

The additions in the Animals and recording procedure section are helpful!

- L 123-127. It's a little confusing that two slightly different age ranges for social maturity are provided in this section - suggest removing one.
- L 135. Closed parenthesis missing.
- L 185. Should read "...previous knowledge of other..."
- L 196. "500 Hz for adults' calls..."
- L 250. Suggest "...number of good-quality, complete calls in the smallest..."
- L 277. Should this be "...considering sequences with barks..."

Results

- L 300. electronic supplementary material
- L 411. Micro-geographic variations in vocalizations
- L 417. Could the potentially louder low-frequency noise at Cape Cross have masked any low-frequency call components?

Discussion

- L 535. Missing closed parenthesis.

Conclusion.

- L 562. Suggest "...therefore are particularly well adapted to the extreme..."

Figures

- Figure 4. Y-axis should be indicator of precision
- Electronic supplementary Audio S1. Suggest verbally annotating this audio file to make it easier for the reader to identify call types and match to Figure 1. This is great to include!
- Electronic supplementary Figure S2. Update labels of age classes to reflect ms terminology (these are no longer used).

Reviewer: 2

Comments to the Author(s)

The authors have adequately revised the manuscript and addressed the comments of both reviewers. One last question: can you add the number of recording days (effort) at both locations in the manuscript? I saw your reply but cannot find the information in the manuscript.

Appendix A

RSOS-202241 "Vocal repertoire, micro-geographic variation and within-species acoustic partitioning in a highly colonial pinniped, the Cape fur seal"

Comments to Author (Dr Ari Friedlaender):

To the Authors,

Both reviewers provide excellent feedback and commentary on your submission. As a first description of the vocalizations of the species there is interest in the results, however, it appears that there are major concerns raised regarding the interpretations of the findings and whether there is justification to be able to place context to certain signals given the data that are presented. I would like the authors to carefully consider the language that is used for interpreting the different acoustic signals described in the submission and what, if any, meaning can be afforded to them. As well, both reviewers found significant and substantial need for major editorial work on sentence structure, language, and clarity. Please consider this carefully and be diligent when revising your submission to be as precise as possible with the text. I am interested to read a revised version of this work and how the reviewer comments are incorporated to more appropriately interpret the findings.

Thank you.

Ari S. Friedlaender

Dear Ari Friedlaender,

Thank you for giving us the opportunity to submit a revised version of my manuscript entitled RSOS-202241 "Vocal repertoire, micro-geographic variation and within-species acoustic partitioning in a highly colonial pinniped, the Cape fur seal". We appreciated the time and effort that you and both reviewers have dedicated to providing us with valuable feedback on this work. We are grateful to both reviewers for their insightful comments.

We have taken into account most of the suggestions provided by reviewers and we have highlighted the changes within the manuscript. Here is a point-by-point response to the reviewers' comments. Apologies for the imperfect numbering of the lines. New numbers have been added and the corresponding line added to each reviewer's comment. The manuscript has been carefully edited for language and clarity - Editing tracks for this aspect are not visible.

Yours sincerely,

Mathilde Martin

Reviewer comments to Author:

Reviewer: 1

Comments to the Author(s)

This manuscript describes the vocal repertoire of the Cape fur seal, which has not previously been reported. Results suggest that this species produces calls that are generally similar to those of other otariid species. Comparisons of recorded vocalizations across call type, age and sex class, and colony demonstrated that certain acoustic features were distinguishable across categories, which may enable discrimination between these groups. This work is important in terms of understanding sound production and communication in a highly colonial marine mammal.

The paper is certainly quite interesting. The manuscript presents valuable data and the recording and analysis methods are generally sound. However, in certain places explanations and corrections are needed (as noted below), and in some cases additional methodological details or clarifications should be provided. The authors should take care not to overextend the interpretations of their results, specifically with regard to the use of these calls for individual or group recognition.

This paper would be improved by careful and thorough editing. I have made note of a few of the typos and grammatical issues, but not nearly all of them. In addition to such errors, many of the sentences are quite long and convoluted, which makes it difficult for the reader to follow. This is particularly true in the Introduction and Discussion sections, which could also be simplified and streamlined for clarity.

Additional, specific feedback is provided below:

Summary

(L3) P 3, L 38-40. This is a strong statement to make for all birds and mammals, as it is certainly not true for all species. It would be more accurate to state that "The acoustic channel is an important modality used by birds and mammals..."

Correction done L3

(L9) P 3, L 48-50. The authors should be careful (here and throughout the ms) to differentiate between calls having the potential to be used for discrimination (among individuals, age or sex classes, etc.) and calls actually being used in that way. The latter would need to be documented through targeted behavioral studies with this species.

Sentence modified: "Intra-species variations in these vocalizations highlight a potential ability to convey information about..."

(L11) P 3, L 51-53. This assertion depends on how "distinct acoustic niches" are defined – just because there are some significant differences in spectral and temporal patterns across call types, this doesn't necessarily mean that the calls occupy distinct acoustic space. Clear evidence needs to be provided (in the main body of the ms) to support this claim.

‘Acoustic niches’ have been defined by Krause (1993): “[...] each creature appears to have its own sonic niche (channel, or space) in the frequency spectrum and/or time slot occupied by no other at that particular moment”

Here, we show that spectral and temporal features are sufficiently different that there is no overlap among call types (Figure 6). This is, in my opinion, in line with the Acoustic Niche Hypothesis from Krause. The definition of ‘acoustic niches’ is now provided (L97-99).

Introduction

(L16) P 4, L 7. Suggest “fundamental” instead of “primordial,” here and elsewhere.

Modifications done here (L16) and in the abstract (L1)

(L21) P 4, L 15. Communication can involve other sensory modalities as well. To be complete, suggest including all of these here, or adding “e.g.,” inside the parentheses.

The parentheses have been removed as the sensory systems are listed above.

(L22-24) P 4, L 18-20. These general statements about acoustic communication are sometimes true, but not always. It is not always true, for example, that acoustic signals propagate over long distances or that they can be easily localized (these depend on the temporal and spectral characteristics of the signal, the environment they’re produced in, and the auditory capabilities of any listeners).

Statement reformulated (L22-24)

(L24-27) P 4, L 20-21. Again, whether sound is the most efficient sensory cue for communication really depends on the environment, the signal of interest, and the sensory capabilities of the intended receiver(s). It is not always true for all species, distances (between signaler and receiver), conditions, etc. This is discussed later in the paragraph in terms of the factors limiting the effectiveness of vocal exchanges. For accuracy, these statements should be revised.

You’re right, thanks - Statement reformulated (L24-27)

(L44) P 4, L 42. What is meant here by encoding-decoding mechanisms? Is this referring to 1) mechanisms of sound production, and 2) the sensory and cognitive processing capabilities of the relevant species? Something else? Suggest explaining more thoroughly or re-phrasing so the reader can follow more easily if unfamiliar with this specific terminology.

We have now added more explanations about the coding-decoding processes (L40-43).

(L47) P 4, L 45-47. Suggest defining taxa that the reader might not be familiar with (e.g., cetacean, phocid) or being more consistent with naming conventions across taxa (e.g., whales or seals could be used to be more parallel).

Done (L47)

(L47-52) P 4, L 47-53; **(L62-65)** P 5, L 8-18; **(L79-83)** P 5, L 33-38; **(L94-104)** P 5-6, L 54-3. Some examples of when writing is a bit dense and difficult to follow.

Sentences were shortened or split (L47-52) and (L62-65)
Paragraph reformulated (L94-104)
Sentence removed because of redundancy (L79-83)

(L54) P 4, L 54-56. Please provide the scientific name of the Cape fur seal. Also suggest defining Otariid, for example with “Otariid (eared seal),” here.

This is now added (L54)

(L56) P 4, L 56-58. Suggest “...because of the extreme density of colonies and their social structure during the breeding season, which require...”

Done (L56)

(L58-60) P 5, L 3-8. Suggest removing these details about Cape Cross, since they’re also included in the Methods section.

Absolutely – Done (L58-60)

(L69) P 5, L 19. Suggest defining pinnipeds here for readers who are unfamiliar.

Done (L69)

(L75) P 5, L 28-29. This should read “...initial investigations on a larger scale and explored...”

Done, Thank you (L75)

(L77-79) P 5, L 31-32. Suggest something like “...we aimed to assess whether the signal contained unique acoustic features which could potentially be used to convey information about....”

Done (L77-79)

(L79-83) P 5, L 33-38. What is the basis for this hypothesis? Has this been shown in other species? Are there relevant studies the authors could refer to here to support this hypothesis?

We decided to remove this sentence as this is not relevant with our first aim, and redundant with the second aim (L79-83).

(L85) P 5, L 41. Suggest deleting “in both study sites” since the study sites have not been introduced yet.

Deleted (L85)

(L89-91) P 5, L 44-47. Suggest framing this more as a challenge or constraint these animals face, rather than something that they certainly do.

Sentence reformulated (L89-91)

(L108) P 6, L 19. It would be helpful to define micro- and macro-geographic variations (here or elsewhere) since these are important concepts in this paper.

Definitions provided (L108-111)

Materials and Methods

(L122) P 6, L 29. In this section, the different sex and age classes of CFS recorded at each colony should be clearly described.

We added two sentences to clarify both age and sex classes (L. 124-134) and (L163-164)

(L122-123) P 6, L 34-35. What is the age range of subadult males? Less than 9 yo and greater than xxx?? Was this determined through observation of any physical characteristics, or based only on whether or not the male formed a harem? What if a male was sexually and socially mature, but did not establish a territory or form a harem this particular year? Please provide more details.

We determined the males' age class (subadult vs. adult) based on their physical characteristics and not on their abilities to hold harems as some adult males do not hold harems. As most fur seals, Cape fur seal males become sexually mature at about 5 years (i.e., subadults), but they only reach the social maturity when they are 8 to 13 years old (Jefferson et al. 2011). Adult males are much bigger than subadult males, and they also show specific physical features: enlarged neck and shoulders, mane with longer guard hair around the neck and shoulders. The details about males' sexual status are now explained (L. 124-134).

(L154) P 7, L 8-14. Since directional microphones were used, were all calls recorded on axis? If not, how might this have affected the measured acoustic features of these calls? Please provide more details here.

All calls were recorded on axis – the microphone was directed to the calling individual to ensure vocalizations of good quality. This is now added (L 158-159).

(L159-160) P 7, L 12-13. How accurately can the sex of pups be determined visually, at a distance? Was age class always determined, or only for males? Were female subadults not recorded, or were they grouped with adults, and why?

- For sex determination in pups, it's explained right after (L170-171): pups were briefly caught to determine their sex. Sexing cannot be performed at distance for pups. Most of them were marked with hair-dye so we knew their identity while recording them.

- Age class was specified only for males because they go through two different stages resulting in different social behaviours: subadult are 5-8 years old, they are socially immature and they don't take part to the reproduction process (i.e. territoriality, mating with females) while adults (older than 8 years) have territory and mate with females. Females are sexually mature at 3-6 years (Wickens and York 1997) so all females older than this age are considered to be adult. Subadult age class is not relevant for females as they won't produce any pup-attraction calls and females included in this work were only adult females with pups.

Clarifications made (L. 124-134) and (L163-164)

(L166+) P 7, L 16+. Was this age estimation of pups based on other studies -- can references be provided? And again, how were older animals categorized? More details are needed.

It's an estimation we based on our observations on pups marked at birth or few days after birth: the time it takes for the umbilical cord to disappear after birth and the appearance of the first signs of moulting (at about 4 months of age, Jefferson et al. 2011).

(L160-162) P 7, L 30-31. Was the recording distance for each call noted or considered in the analysis?

Propagation through the colony could have modified the frequency characteristics of the recorded vocals, which would have an important impact on the results of this study.

The recording distance was not noted for each call because all our recordings were performed within 6 meters as stated at L160-162, limiting the difference in recording quality and potential modifications on the acoustic characteristics of the calls.

(L156-158) P 7, L 28-29. Did the experimenters typically observe any behavioral disturbance upon their arrival? Was this 15-minute acclimation period after the experimenter(s) arrived at the colony, or after behavior returned to baseline?

Cape Cross is a touristic area so CFS are used to human presence and the arrival of experimenters do not induce any behavioural disturbance. At Pelican Point, seals are more skittish when humans approach the colony. The experimenters made very slow and controlled approaches by crawling in the sand and by stopping if any disturbance was observed. The approach time to reach the recording distance without causing disturbance to the seals was between 15 to 20 minutes. Then, once in position, a period of 15 minutes was waited, to ensure a baseline behaviour, which usually took much less than 15 min.

(L180) P 7, L 42-43. Please provide frequency resolution with these settings and sampling rate (i.e., 3-dB filter bandwidth).

The sampling rate is already given (L177), and we have now added the frequency resolution (L180-181).

(L181) P 7, L 42-43. How were “good quality” calls defined? Was it just based on SNR? Were incomplete (partially recorded) calls considered in this analysis.

In this study, the quality of a call on a recording is related to low background noise (good signal-to-noise ratio, see L181-182) and the absence of overlap with other vocalisations (L181-182). Given the duration of the calls, the only reason that a call is incomplete is if it overlaps with other vocalizations, in which case the call was not included in the study. Therefore, all the calls considered are complete, non-overlapped and have a sufficient SNR so that the energy spectrum is not modified by the background noise or the propagation degradation.

(L181) P 7, L 44-45. Can “low background noise” be quantified? Was there an SNR cutoff? How much did ambient noise conditions vary day to day, and did they vary significantly across colonies? Providing information about the noise background at each colony (or at least more quantitative information about acceptable SNRs) is necessary to ensure that this did not have any effect on the results.

We did not use any SNR cutoff, as the most important aspect for us was to avoid any overlap between recorded vocalizations. We are working in a colonial environment where silences are rare so that recording ambient noise without any seal vocalizations is difficult. Saying this, from our analyzed vocalizations, the SNRs were above 15 dB the background noise (ambient plus non focal seal calls). Moreover, all individuals were recorded in similar conditions (within 6 meters) and this is also why we used a directional microphone, so that we could target one individual in particular and get the best SNR. The ambient noise (non seal generated) varies within and between days, depending on the weather (more and less hot), wind, swell, and also the presence of birds around the colony.

(L183-184) P 7, L 47-48. Why was 100 Hz chosen for the high-pass filter cutoff? According to Table 1,

multiple call types have fundamental frequencies close to 100 Hz. In fact, growls produced by females at Pelican Point had a mean fundamental frequency of 90 Hz. This seems problematic for this dataset, and should be addressed or corrected.

- The high pass filter was set at 100Hz because most of the background noise was between 0 and 100Hz and thus not filtering the noise would greatly alter the energy distribution within the vocalization spectrum.

- The frequency values of the fundamental were calculated from the spectrum (harmonic cursor can be used in Avisoft to detect the fundamental frequency using frequency interval between harmonics - this is added now L192-193) as most of the time (and also in many mammalian vocalisations), the fundamental is filtered by the vocal tract, so it won't appear on the energy spectrum. Based on this, the high-pass filter is not an issue for the measurement of the fundamental frequency.

(L188) P 7, L 49. Were calls categorized into types based on common perceptual features, or based on context? Please clarify. The way this is written, it sounds like context was the main driver for call classification.

Indeed, our classification was based on the social context as call types have been already defined in other otariid species, and thus, we also based this classification on the previous knowledge. We added this information (L. 188-190).

(L196) P 7, L 60. There is a period missing at the end of the last full sentence on this page.
Yes thank you, done (L196)

(L199) P 8, L 6-7. Again, frequency resolution (3-dB bandwidth) with these settings would be useful to provide here.

Now provided (L199)

(205) P 8, L 15-16. This is a little unclear. These additional variables were specific to a given call type AND were not always measurable for all calls of that type? Or should that be "or" there?

It's a 'AND': Additional variables were specific to a call type and couldn't be measured on all calls of this type. For example, some females produce a fast amplitude modulation in their PAC: acoustic parameters such as AMprop and AMrate are thus relative to this amplitude modulation and thus these features were specific to PAC only (no AM in barks or FAC for instance) AND they were not found in all PACs because not all females produce these fast amplitude modulations. Sentence modified (L206-207).

(L212) P 8, L 23-34. Can bleating/quavering be defined here or elsewhere? A reference is provided, but this would still be helpful.

We have now added the definition for bleating/quavering (L212) (i.e. fast frequency modulation)

(L215) P 8, L 28-29. It's confusing here, and elsewhere throughout the Methods, that the call types are mentioned/alluded to but not defined. Suggest defining the five main call types in the Methods if they're going to be discussed at all.

The five call types are now mentioned (L189)

(L215-216) P 8, L 30. Were soft versus aggressive growls context specific, or was the only difference the amplitude of the calls? It's a bit misleading that one name refers to (relative) amplitude and the other refers to context, and that that two are mutually exclusive. Were "soft growls" ever produced aggressively? How loud did the call have to be to be considered soft versus aggressive? Please provide more explanation.

We agree that at this stage, this may be confusing for the reader. We have removed the two subcategories as these are well described in the result section and we use now 'soft' growls (produced during minor conflicts) and 'loud' growls (produced during hostile interactions, generally associated with physical demonstration of aggressiveness).

(L229) P 8, L 52-53. Can a reference be provided for the Random Forest algorithm?
Added Breiman 2001 (L229)

(L246) P 9, L 36. FAC has not been defined yet.

We have now defined PAC and FAC in the method section (L189).

(L244) P 9, L 56. Up until now, the methods have seemed to focus on pups. It was not completely clear before this point whether adults were actually recorded. This needs to be specified earlier (which age and sex classes were recorded and, again, how they were defined).

We now added a sentence in 3.3.1 (L. 163-166) and clarified in (L245) and (L251-254)

(L268) P 9, L 58. Again, please make sure that distinctions between age classes (if any) are clearly described for both males and females.

'Male' added everywhere (L268) (L275-276)

(L273) P 10, L 11-13. Adults and subadults in these comparisons are presumably all males? Please clarify.

This is now clarified (L273-276)

(L284) P 10, L 26-27. These calls types have not been identified or described yet. These abbreviations have not been defined, so should not be used here.

Calls types have been now described earlier (L189) with their abbreviations, but we have put their names in full here for clarity (L. 284).

Results

(L297) P 10, L 49. Do the males and females listed here represent all age classes (aside from pups)?
Details have been added now (L297)

(L301) P 10, L 50. It would be useful to provide the relative proportions of each call type in the Results (either in the main text or in Table 1).

We have now added the proportions for each call type (L301)

(L309-310+) P 11, L 9-10. What are these values? Previously the authors stated that mean and stdev were used, so why is a range provided here? Based on Table 1, I believe these are the means for the two colonies, and the standard deviations for the two colonies. This must be explicitly stated, especially since this is not standard notation. Furthermore, the data from the two colonies are not always presented in the same order, so this should be noted as well (data are provided from low to high instead).

Here we presented the mean values for both colonies with their SD values. We have now added this information in the text (L310 and further).

(319-320) P 11, L 24-25. How did sample size compare for the two call types? Could this have affected their relative standard deviations?

We have a total of 865 female attraction calls and 766 pup attraction calls so the sample sizes are relatively similar. For all parameters (except the duration), the standard deviations values for pups' calls are most of the time twice as big compared to SD values for females' calls. To our opinion, this highlights a significant variability in pups calls compare to females' ones.

(L326) P 11, L 3. Why were some of the prediction plots included as supplements and some as part of the main text? Could these be combined into a multi-panel figure potentially?

Since this ms is already very long and dense, we have chosen to present the most relevant plots, and to include as supplements those that could be explained in the main text.

(L335-337) P 11, L 49. The pup age classes were described in the Methods, but I don't believe they were explicitly assigned numbers (age classes 1 through 3). Please confirm and define in the Methods if necessary.

You're right, thank you for pointing this out. Age 1, 2, 3 are now removed to keep the age-classes: less than 2 weeks, 1 month and 2-4 months (L335-337,342).

(L336) and it's not obvious that age classes 1 and 3 are "quite distinct" acoustically.

On the DFA plots, we agree that calls from age class '< 2 weeks' overlap with calls from other age classes, and the most striking difference is between age classes '1 month' and '2-4 months', and not '< 2 weeks' and '2-4 months', so we have now changed this. However, the statistical analyses revealed significant differences between age classes for many features (see Table 2), including differences between age classes '< 2 weeks' and '2-4 months. Changes made (L336-337)

(L363) P 12, L 31-32. Should read, "...accuracy of prediction values were 83% (CC) and 78% (PP)..." The authors should check tenses and number agreement (between nouns and verbs) throughout the manuscript for accuracy. There are a lot of issues with this.

Thank you, correction done + manuscript re-checked (L363)

(L362) P 12, L 47-48. To clarify, were subadult versus adult females not separated by age class? Presumably not, but this is not explicitly stated anywhere. Also, in some places "adult females" is specified. It is unclear whether subadult and adult females were grouped together, or if subadult females were not considered. These details need to be included, clearly, in the Methods.

The clarifications added in the Methods 3.1 (L124-134 and L163-164) about the ages classes of males and females address this point.

(L370-372) P 13, L 43-47. Suggest revising and splitting into two sentences.

We split the sentence in two as suggested (L370-372).

(L373-379) P 13, L 50. Should this be “Micro-geographic variation in vocalizations”

These results are not part of the micro-geographic variations. Within each study site (PP and CC) we compared the acoustic characteristics of barks. At PP, this comparison was conducted between the barks of adult males, subadult males and adult females. At CC this comparison was conducted between subadult males and adult females. No comparison of the acoustic features of barks between sites was conducted.

(L373-379) P 13, L 52-55. Why were these particular call type/sex/age class combinations chosen for comparison? Are these adult females or all females?

No subadult females were studied (this age class does not exist for females actually), only adult females. We have now added this information (L252-254 and L267-268) + (L131-134). So we compare adult females with subadult males on one hand and adult females with adult males on the other hand. In addition, comparisons between adult and subadult males were also performed.

Discussion

(L426) P 14, L 18-19. Suggest “...overall structure and biological function are apparently similar...”

Changed done as suggested (L426)

(L430) P 14, L 25. Fond should be “find” in this sentence.

Thank you, change done (L430)

(L432) P 14, L 29-30. What was the sex ratio at each colony? Data should be provided in the Results to support this claim.

We can't answer this question as these data are not available for the study colonies and such ratio is always difficult to assess in otariids as adults are often at sea to forage. In the literature, it is common to consider for otariid species an average adult sex ratios of up to 10 females per male (Gentry 2008, Encyclopedia of marine mammals, Academic Press, p. 339-342).

The reported harem sizes for Cape fur seals are much larger (average of 30 females per male with a maximum of 66 females – Shaughnessy 1978) than other fur seal species (5 to 10 females, Riedman 1990). For this reason, we considered that males have more females available than other fur seal species. This information has now been added (L432-434).

(L446) P 14, L 48-49. Suggest replacing “genders” with “sexes.”

Change done (L446)

(L448) P 14, L 52. Can the estimated divergence time be provided (X mya)?

Information is now provided (L448)

(L448-449) P 14, L 52. Or, were these calls added to the AFS repertoire later? Unless we know something about vocal behavior in their most recent common ancestor, this can't really be determined here. This statement should be revised.

Sentence has been modified (L449)

(L469-471) P 16, L 24-27. There are multiple colons in this sentence, suggest re-organizing.

Done (L469-471)

(L521) P 16, L 41-49. While the call types have some recognizable (and unique) acoustic features, it is not clear that this constitutes a “distinct niche” for each call type. There is still overlap between types, and across age and sex classes. Are the authors referring to individual recognition here (L 46)? Because these data do not speak to that. Importantly, just because there are differences in acoustic features across call types, this does not necessarily mean that receivers are utilizing this information for individual, sex, or age class recognition. That would need to be further investigated throughout behavioral studies (e.g., playback experiments). The authors should be careful not to overextend the interpretation of their results.

Here, we are discussing the fact that call types are well separated, even if there is some overlap between some call types (FAC and PAC for instance), the overlap is quite limited as shown by Figure 6. So this means that Cape fur seals may be able to discriminate among call types. We are not referring to individual recognition as this is the topic of another publication (Martin et al., in press). Here, we are simply highlighting the fact (supported by our statistical analysis) that each call type has specific features which can facilitate call discrimination in a busy colony.

Based on this, we have changed one sentence that might have induced this confusion, see (L521-523): In a group, the more members interact, the more they benefit from recognizing *the call type a sender is emitting as it gives crucial information on the type of social interaction (i.e., agonistic or affiliative)*.

(L526) P 16, L 51. Suggest “...CFS vocalizations contain information...” instead of “transmit information.” While there might be differences across calls that provide information, it is unclear the extent to which that information is actually used by receivers in this species.

Change done (L526). We agree with referee 1 that at this stage, we don't know which information is used by receivers. Only playback experiments will reveal the use of such information encoded in calls.

(L542) P 17, L 35. This is certainly a possibility, but are the individuals at Cape Cross actually smaller, or is there reason to suspect that this may be the case?

Unfortunately, we don't have any information about the body size distributions of the CFS colonies we have studied. We requested this from the Ministry of Fisheries and Marine Resources in from Namibia, but we did not get any answer. This statement is therefore purely a hypothesis. Possible

reasons for differences in body size would be different levels of food resources between the two areas, or genetic variations - which can only be validated through further genetic research.

(L545) P 17, L 23-26. Are ambient noise levels relatively higher at Cape Cross, or concentrated at low frequencies? This is another reason to discuss ambient noise levels in the Methods or Results.

We haven't measured the ambient noise levels in these two colonies, but there are clearly different habitat characteristics. Pelican Point is a sandy Peninsula; and the area where recordings took place was sheltered from sea swell. In contrast, Cape Cross is a rocky area exposed directly to the swell which generates wave noise at low frequencies (30-500 Hz). This is why we suggested that differences in habitat/environment could be a potential explanation for the differences in frequency content between the two colonies. We have now added these differences in wave noise in the Methods section (L146-148 and L152-153).

(559-560) P 17, L 48-49. Not really. This would just facilitate discrimination among age and sex classes (not individuals).

We have modified the sentence accordingly (L. 559-561: Variations between age- and sex-classes in calls facilitate discrimination among animals of different social classes and thus enhance social interactions in a noisy and confusing environment)

(562) P 17, L 50-51. Is it really true that the vocal repertoire of CFS is “particularly adapted to the extreme colony density” – how does the repertoire compare to a related species with less dense colonies?

I think our sentence is not fully correct, as this is not the vocal repertoire that is well adapted to the extreme colony density, these are the acoustic characteristics of each call type. So, we have changed the sentence (L. 561-563).

Tables

Table 1 caption. For the significance code, 5 significance values are provided but only 4 labels are given. Please check for accuracy (here and elsewhere). Also suggest a more explicit explanatory sentence. What does NA refer to in this context? Suggest replacing “brackets” with “parentheses” in caption.

*Significance code 0 '***' 0.001 '**' 0.01 '*' 0.05 'NS' 1*

One label corresponds to a range of values: *** for p-values between 0 and 0.001, ** for p-values between 0.001 and 0.01 and so on...

Table 1. Are these females (for barks and growls) of any age?

Yes, only one age class for females = adult females

Table 2. What age classes are between 4 months (pups) and adult females? The age structure used in this study needs to be more clearly described.

Between pup and adult females = yearling (1year old pup) then juvenile (about 2-4 years) but none of these age classes were considered in this study. We have defined the different types of individuals recorded in this study in the methods section.

Figures

Figure 1. Please include spectrogram parameters in the caption. Caption for panel b: these are presumably provided in age order from left to right, but please clarify. To facilitate comparison, suggest plotting the different call types on common time and frequency axes – otherwise, note in the caption that they are plotted on different axes.

We have modified the caption accordingly.

Figure 2. Are these data pooled across age and sex classes and behavioral contexts? Details such as this should be provided in the figure captions so the reader doesn't have to refer back to the main text.

Clarification made

Figure 3. In the caption, suggest including the variables that contributed most to the total variation, for reference.

Done

Figure 6. This figure is much more convincing than the other LDA scatterplots, which show a lot more overlap between call types/age or sex classes (e.g., Figure 5).

We fully agree. This is exactly what we want to point out in this paper. On one hand the five call types we described are easily distinguishable because their acoustic parameters are very different i.e. they are well separated in their acoustic space. On the other hand, finer variations occur at smaller scale – within the same call type – between age- or sex-classes.

In Figure 3 we show that pup calls slightly differ with age and vary gradually as they grow up. There is overlap between the age groups which shows a form of transition between age classes. See in Results (L335) “The LDA scatterplots (figure 3) showed that, especially for PP, age-classes “1 month old” and “2-4 months old” were the most distinct, with age-class “less than 2 weeks old” sitting between them. This might suggest a progressive trend with age: when getting older, pups increased the duration of their calls, and the distribution of energy was more evenly distributed among harmonics (figure 1.c; table 2).”

In Figure 5 we show that barks produced by adult males and adult females are more different than those produced by subadult males that sit between. The overlap is due to the fact that they are all barks, so short calls produced in series. However, fine acoustic variations allow to separate them.

Reviewer: 2

Comments to the Author(s)

RSOS-202241

Title: Vocal repertoire, micro-geographic variation and within-species acoustic partitioning in a highly colonial pinniped, the Cape fur seal.

This manuscript is interesting and provides proper information about the in air vocal repertoire of the Cape fur seal, as well as micro-geographic variation and within-species acoustic partitioning in this highly colonial pinniped species. In my opinion, parts of the description of the context, and methods could do with a little more detail or explanation, but otherwise I think the manuscript is highly valuable.

Note to the authors: Please next time use actual line numbers for an easier review.

Main comments: I found some background material to be lacking in the introduction/methods – I did not have a good sense of the two locations, the density of Cape fur seal in the areas, and I did not have a clear sense of the animals that would be in acoustic or visual range.

Please provide scientific names for each species [for example Page 4: South American fur seal [47,48], Australian fur seal [49], Subantarctic fur seal [50] and Northern fur seal [51].]

Scientific names are now provided.

Specific recommendations are provided below:

ABSTRACT

(L8) “We described the acoustic features and social function of five IN AIR call types...”

Clarification made (L8)

INTRODUCTION

(L72) Page 2: Second paragraph: You should mention that your first goal focus during the breeding season.

We have now added this information (L72)

(L86-88) Page 2: third paragraph : “... it is an extremely noisy environment combining a high risk of vocal and visual confusion among conspecifics.”. Visual? Do you mean “auditory” ?– A noisy environment should not affect the visual perception. Also, what do you mean by vocal confusion? Do you mean that the animal does not know which sound produce? I would suggest rephrasing this sentence.

We agree that visual is not appropriate here and the visual aspect is not the point – We have now replaced by “auditory confusion” (L86-88)

Page 2: third paragraph: Does different group age, sex and social status are mixed/grouped in the

colony?

Yes, we have now added information (L85-86)

MATERIAL AND METHODS

(L122+) Page 3: 3.1 First paragraph – I would suggest moving this paragraph in the introduction.

We prefer to leave this first paragraph in the methods section, as it gives very specific information on our two study sites. Moreover, referee 1 asked us to add in additional information and now this paragraph is much longer (nearly 10 lines longer than in the first submission). So, we decided to leave it here.

(L140+) Page 3: 3.1 Second paragraph – What is the distance (km) between sites? Do Cape fur seal stay all year round near to their colonies or can they switch from one colony to the other? How many Cape fur seal colonies do you have along the Namibian coast? I would suggest adding a figure with a map showing both locations (and potentially some other colonies) and including some pictures of each location to provide a better background to the reader.

You said that the breeding is a 4 month period, how many times did you go in the field for the recordings? Maybe provide the number of days that you spent to each location?

The two colonies are separated by 150 km (see L106 and L153).

There is limited information in the movement of individuals between the two study colonies (Pelican Point and Cape Cross) and their use of breeding colonies (i.e. potential fidelity to a birth or breeding site). Cape fur seals have been shown to undertake foraging trips of several hundred kilometres and individuals can cover a fairly large area. Few individuals that have been tagged at Cape Cross have been sighted to feed off the coast of Pelican Point (Skern-Mauritzen, M., Kirkman, S. P., Olsen, E., Bjørge, A., Drapeau, L., Mejer, M. A., ... & Oosthuizen, W. H. (2009). Do inter-colony differences in Cape fur seal foraging behaviour reflect large-scale changes in the northern Benguela ecosystem?. *African Journal of Marine Science*, 31(3), 399-408.). The reverse is likely to occur. So there might be some movements of individuals between these two breeding colonies separated by only 150 km (a short distance for a fur seal) but we do not have empirical data to report here.

We stayed in the field for the 4 months (from mid November to early March) and spent 60 days at Pelican Point and 5 days at Cape Cross for recordings.

We have now added a figure showing the two colonies on a map and a picture showing the different habitats (figure in supplementary).

(L182-183) Page 4: 3.2 - “A maximum of 10 calls per individual were included in the analysis”. Is it a combination of call types or only one call type? (10 of the same call type or any call type?) “Signals were categorized into different call types based on the behavioural context of production...” Please provide the call types here. Later in the paragraph and the next one, you mention barks, pup-attraction calls and growl.

Call types are now provided (L. 189) and we also added clarification regarding the 10 calls per individuals of the same call type (L190-191).

(L200-201) Page 5: “..., measurements were thus performed on 5 barks randomly chosen ...” – Does the 5 barks are included in the maximum of 10 calls?

Clarification is made in the earlier sentence (190). For barks, we chose a maximum of 10 bark sequences, and thus for each sequence, we chose 5 barks.

(L272-273) Page 9: “trees were built with the same sample size for each group (i.e. the size of the smallest group): n=135 for Cape Cross and n=150 for Pelican Point. Does n is the number of individuals or the number of recorded barks?”

True, it's the number of barks, clarification made (L272-273)

RESULTS

(L345-346) Page 9 – “Differences between sexes were only investigated for very young pups at Pelican Point”. Why not for older pups?

Explanations are now given (L346). Very young pups were briefly caught (when possible), marked and their sex was determined. For older pups which are harder to catch, marking was conducted using a pole to apply the dye. As we did not catch these older pups, and pups not sexually dimorphic, we could not assign the sex.

DISCUSSION

(L441-442) Page 11: “The role of this call remains unknown but could be related to a high level of stress or emotional state”. Speculative. Please provide more justification (why animals would be stressed...etc) or delete it.

As we don't know the potential role or function of this call, we prefer to remove the second part of our sentence (L. 441). Further investigations are needed to provide a better justification.

(L443-444) Page 11: “Cape fur seal and Australian fur seal ... are two sub-species of *A. pusillus*”

Change done (L443-444)

FIGURE

Figure 1. Please provide arguments used for the spectrogram (window, overlap, fft). Also I would suggest to include some audio files corresponding to each CFS call type as supplemental material.

We now added information about the spectrogram settings, and we will also provide sound samples for each spectrogram as supplementary material.

Figure 5: adult male and females and subadult males

Changes done